# Inherited salt-losing tubulopathies are associated with immunodeficiency due to impaired IL-17 responses

Rhys D. R. Evans [1✉], Marilina Antonelou[1], Sanchutha Sathiananthamoorthy[1], Marilena Rega[2], Scott Henderson[1], Lourdes Ceron-Gutierrez[3], Gabriela Barcenas-Morales[4], Christoph A. Müller [5,6], Rainer Doffinger[3,7], Stephen B. Walsh [1,8✉] & Alan D. Salama [1,8✉]

Increased extracellular sodium activates Th17 cells, which provide protection from bacterial and fungal infections. Whilst high salt diets have been shown to worsen autoimmune disease, the immunological consequences of clinical salt depletion are unknown. Here, we investigate immunity in patients with inherited salt-losing tubulopathies (SLT). Forty-seven genotyped SLT patients (with Bartter, Gitelman or EAST Syndromes) are recruited. Clinical features of dysregulated immunity are recorded with a standardised questionnaire and immunological investigations of IL-17 responsiveness undertaken. The effects of altering extracellular ionic concentrations on immune responses are then assessed. Patients are hypokalaemic and hypomagnesaemic, with reduced interstitial sodium stores determined by $^{23}$Na-magnetic resonance imaging. SLT patients report increased mucosal infections and allergic disease compared to age-matched controls. Aligned with their clinical phenotype, SLT patients have an increased ratio of Th2:Th17 cells. SLT Th17 and Tc17 polarisation is reduced in vitro, yet STAT1 and STAT3 phosphorylation and calcium flux following T cell activation are unaffected. In control cells, the addition of extracellular sodium ($+40$ mM), potassium ($+2$ mM), or magnesium ($+1$ mM) reduces Th2:Th17 ratio and augments Th17 polarisation. Our results thus show that the ionic environment typical in SLT impairs IL-17 immunity, but the intra-cellular pathways that mediate salt-driven Th17 polarisation are intact and in vitro IL-17 responses can be reinvigorated by increasing extracellular sodium concentration. Whether better correction of extracellular ions can rescue the immunophenotype in vivo in SLT patients remains unknown.

[1] Department of Renal Medicine, University College London, Royal Free Hospital, London, UK. [2] Institute of Nuclear Medicine, University College London, University College London Hospital, London, UK. [3] Department of Clinical Biochemistry and Immunology, Addenbrookes's Hospital, Cambridge, UK. [4] Laboratorio de Inmunologia, FES-Cuautitlan, UNAM, Mexico. [5] Department of Radiology, Medical Physics, Medical Center University of Freiburg, Faculty of Medicine, University of Freiburg, 79106 Freiburg, Germany. [6] German Consortium for Translational Cancer Research (DKTK), Partner site Freiburg, German Center for Cancer Research (DKFZ), 69120 Heidelberg, Germany. [7] National Institute of Health Research (NIHR), Cambridge Biomedical Research Centre, Cambridge, UK. [8] These authors contributed equally: Stephen B. Walsh, Alan D. Salama. ✉email: rhys.evans@ucl.ac.uk; stephen.walsh@ucl.ac.uk; a.salama@ucl.ac.uk

Globally, dietary salt (sodium chloride) intake exceeds the amount that we consumed in our relatively recent evolutionary past, and in most communities is greater than the World Health Organisation recommended limit of 2 g sodium per day[1,2]. Sodium intake is linked to the development of hypertension and cardiovascular disease, but other adverse effects of excess sodium are beginning to emerge. These include an increasing awareness that sodium directly impacts immunity and is associated with the development of inflammatory and autoimmune disease[3,4].

IL-17 is a pro-inflammatory cytokine produced by CD4+ (Th17) and CD8+ (Tc17) T cells. IL-17 responses provide protection from infection, particularly at epithelial surfaces, but are increasingly implicated in autoimmune disease[5]. Inherited defects in IL-17 immunity lead to chronic mucocutaneous candidiasis (CMC), which is characterised by mucosal bacterial and fungal infection, often associated with other clinical features of dysregulated immunity including increased allergic disease. CMC is most commonly due to inherited mutations affecting the phosphorylation and activity of the cytokine signalling molecules Signal transducer and activator of transcription 1 and 3 (STAT1 and STAT3), which subsequently impacts Th17 polarisation[6,7].

Chronic salt loading leads to sodium storage in muscles and the skin, resulting in interstitial sodium concentrations at these sites in excess of those in plasma[8]. Sodium here may be cleared by macrophages and this finding provides the initial evidence directly linking extracellular sodium to immune cell function[9]. Sodium has since been shown to impact multiple components of both innate and adaptive immunity[4,10]. Increased sodium promotes activation of pro-inflammatory cell subtypes such as Th17 cells and M1 macrophages, while inhibiting regulatory cells[11–14]. Intracellular pathways that mediate this inflammatory effect of sodium are beginning to emerge. In Th17 cells, this involves upregulation of nuclear factor of activated T cells 5 (NFAT5) and serum/glucocorticoid-regulated kinase 1 (SGK1)[12,13]. The consequences of salt-driven activation of these inflammatory cells are increased autoimmunity, exacerbated transplant rejection, and increased clearance of cutaneous infection[11–13,15–18].

Renal sodium handling involves the glomerular filtration of large amounts of sodium, which is then reabsorbed across the tubular epithelium, a process that is homeostatically controlled mainly in the distal nephron. This involves a number of different sodium transporters often functioning in concert with other ion transport mechanisms[19]. Inherited defects in renal sodium reabsorption due to mutations in genes encoding sodium or other ion transporters underlie the salt-losing tubulopathies (SLT)[20]. The commonest causes of SLT are defects in sodium reabsorption in the thick ascending limb (TAL) and distal convoluted tubule (DCT), leading to Bartter syndrome (BS) and Gitelman syndrome (GS), respectively. Defective reabsorption in the DCT is also a feature of the rarer EAST (Epilepsy, Ataxia, Sensorineural deafness, and Tubulopathy) Syndrome, which has a tubular phenotype indistinguishable from GS[21]. Together, these Salt-Losing Tubulopathies share a number of clinical and biochemical features including chronic salt depletion from renal salt wasting, in addition to hyperaldosteronism and hypokalaemic metabolic alkalosis, commonly associated with renal magnesium wasting and hypomagnesaemia. Patients often report a number of somewhat unexplained symptoms, including recurrent febrile episodes in childhood; a link to autoimmunity in both BS and GS has been described, but not fully explained[22–28]. Immune reactivity in SLT has not been previously explored.

To understand the consequences of chronic in vivo salt depletion, we investigate immunity in patients with SLT. We demonstrate that SLT is associated with clinical features of immunodeficiency due to reduced IL-17 responses. We provide data that suggest the immunophenotype in SLT is a consequence of chronic salt depletion and the associated changes in ionic environment in vivo, and highlight the important role of sodium in addition to other extracellular ions on immune cell activation states. In doing so, we describe a novel cause of IL-17 immunodeficiency and provide the basis for further study of altering extracellular ions to modify infection risk and as a potential immunosuppressive strategy in immune-mediated disease.

## Results

**SLT cohort.** We investigate immunity in 47 patients with genotyped SLT and 46 age-matched healthy and disease controls[29,30]. 23 (49%) patients have BS, 22 (47%) patients have GS, and 2 (4%) patients have EAST syndrome (Supplemental Table 1). Disease controls are patients attending tubular disorders clinics with a diagnosis that is not associated with renal salt wasting (Supplemental Table 2). The median age of SLT patients is 35 (28–43) years and 26 (55.3%) patients are female. Biochemical findings at the time of recruitment are typical of SLT, with hypokalaemic metabolic alkalosis and frequent hypomagnesaemia (Supplemental Table 3). These biochemical features are not seen in controls (Supplemental Table 4). Renin or aldosterone is elevated in 24 (96%) of the 25 SLT patients in whom they are measured.

Total body salt depletion is suggested in SLT by low/normal blood pressure; mean systolic and diastolic blood pressures are $117 \pm 12$ and $74 \pm 7.4$ mmHg, respectively. Six SLT patients underwent $^{23}$Na-MRI imaging of the lower limb, which confirms reduced sodium stores in the skin compared to age-matched healthy controls (HCs) (Fig. 1).

**SLT patients have clinical features of immunodeficiency.** We compare clinical features of dysregulated immunity in SLT patients to healthy ($n = 24$) and disease (non-salt-losing tubular disease; $n = 22$) controls (Table 1). SLT patients have increased bacterial (skin abscess 21%, recurrent upper respiratory tract infection 49% and urinary tract infection 36%) and fungal (recurrent mucocutaneous candidiasis 38%; Supplemental Fig. 1) infections. In addition, SLT patients have an increased prevalence of eczema/dermatitis and allergic disease to a variety of antigens. Moreover, autoimmune phenomena are reported in 14 (29.8%) SLT patients: psoriasis $n = 5$; hypothyroidism $n = 2$; autoimmune hepatitis $n = 1$; vitiligo $n = 2$; Raynauds $n = 2$; and sicca symptoms/Sjögren's syndrome $n = 2$.

Reduced IL-17 responses underlie CMC, which is commonly due to mutations in STAT1 or STAT3 and may be part of the Hyper-IgE syndrome. To assess whether the infections suffered by SLT patients are consistent with an impairment in IL-17 immunity, we devise a scoring system based on the infection-related components of the Hyper-IgE syndrome diagnostic criteria, and apply this to SLT patients and controls (Supplemental Table 5)[31]. IL-17-related infection scores are higher in SLT patients than both control groups (Fig. 2a); moreover, the clinical phenotype of SLT patients is similar to patients with STAT1 and STAT3-mediated IL-17 defects (Supplemental Table 6).

**SLT patients have an altered urinary microbial community.** Given the increased prevalence of SLT mucosal microbial infections, we were interested to understand if there were differences in the microbial community in SLT patients at one of the common sites of infection. Hence, we perform urine sediment cultures, which demonstrate there are altered bacterial constituents with an increase in the presence of *Corynebacterium* sp. in SLT patients compared to controls (Supplemental Tables 7 and 8). To determine whether alterations in extracellular sodium

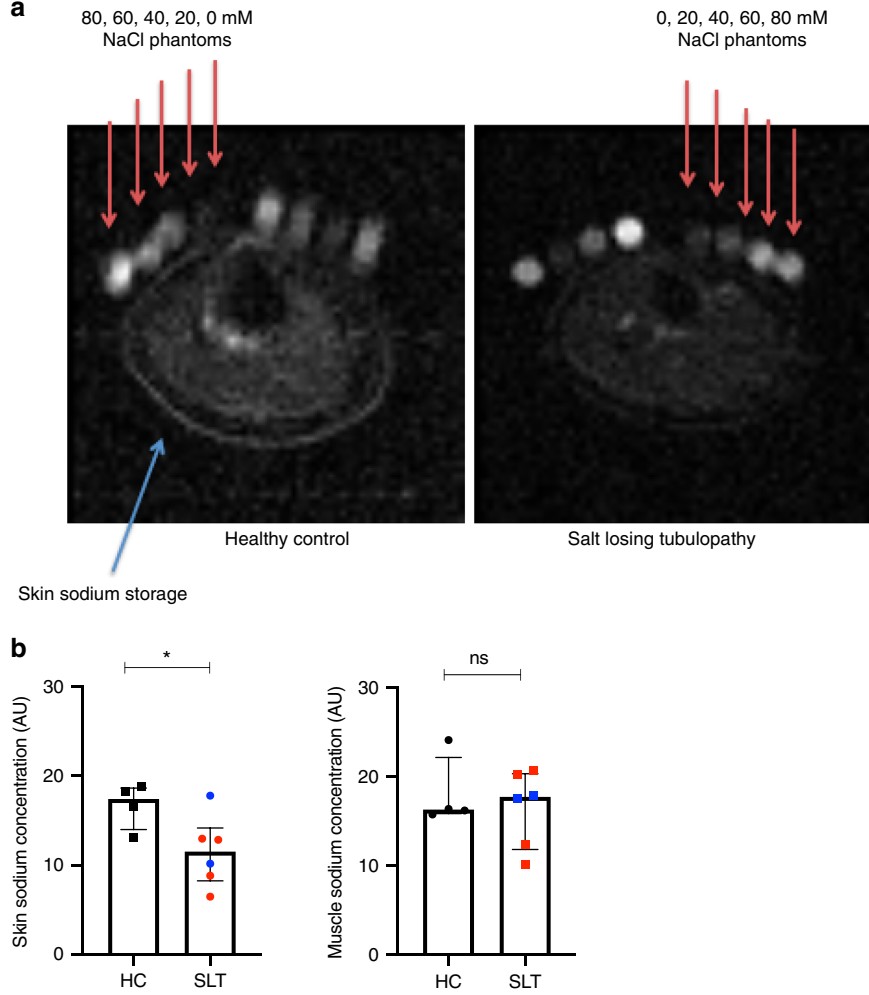

**Fig. 1 $^{23}$Na MRI demonstrating reduced skin sodium stores in salt-losing tubulopathy patients compared to age-matched healthy controls. a** Representative $^{23}$Na MRI images of the lower limb from a SLT patient (Gitelman syndrome) and a HC. Phantoms of known sodium concentration (0–80 mM) are imaged alongside the lower limb (red arrows). **b** Skin and muscle sodium concentrations as determined by $^{23}$Na MRI in SLT patients (GS = 4, red dots; BS type 1 = 2, blue dots) and age-matched healthy controls (n = 4). Skin sodium: HC 17.4 (14.0–18.6) AU, SLT 11.5 (8.2–14.1) AU, p = 0.038; muscle sodium: HC 16.3 (15.9–22.1) AU, SLT 17.7 (11.8–20.3) AU, p = 0.99. Groups are compared with a two-sided Mann–Whitney test. Error bars represent interquartile range around the median. ns not significant (p > 0.05), *p ≤ 0.05, **p ≤ 0.01, ***p ≤ 0.001, ****p ≤ 0.0001. HC healthy control, SLT salt-losing tubulopathy. Source data are provided as a Source data file.

concentration directly affect microbial growth, strains of *C. amycolatum* as well as other commonly encountered microbes (*E. coli*, *S. aureus*, and *C. albicans*) are cultured in media supplemented with NaCl. NaCl has no effect on the growth of the bacteria tested, whereas there is a dose-dependent reduction in the size of *Candida* colonies with additional NaCl (Supplemental Fig. 2). A direct effect of altered extracellular sodium may, therefore, contribute to mucosal fungal infection; however, the lack of an effect of an altered extracellular environment on bacterial growth supports a primary defect in immunity underlying the altered microbiome and increased urinary infection rate observed in SLT.

**Initial immunological analysis of SLT patients**. Immunological analysis of SLT patients is summarised in Supplemental Table 9. Total IgE level is elevated in 10 (33.3%) of 30 patients in whom it was measured, and CD16$^+$CD56$^+$NK (natural killer) cells are reduced in 13 (40.6%) of 32 patients (Fig. 2b, c). Otherwise, parameters tested are within the laboratory reference range.

T-cell proliferation and activation, determined as the percentage of CD4$^+$ cells expressing Ki67 and CD25, respectively, after 72 h of anti-CD3 and anti-CD28 stimulation, are no different in SLT patients compared to controls (Fig. 2d, e). Analysis of antigen-specific antibody responses demonstrates SLT patients have protective levels of varicella zoster and pneumococcal immunoglobulin G, similar to the healthy adult population (Fig. 2f). Moreover, CD4, CD8, CD45RA, and CD45RO subsets are not different in SLT patients compared to controls (Supplemental Fig. 3). Initial immunological analysis of SLT patients, therefore, demonstrates no gross defects in adaptive immunity.

**SLT patients have reduced IL-17 responses**. Given the clinical findings in SLT of increased mucosal infection, and the known effect of salt on Th17 responses, we assessed CD4$^+$ T cell subsets in SLT and compare these to healthy and disease controls. Analysis of IFNγ, IL-4, and IL-17 expression in CD4$^+$ cells, representative of Th1, Th2, and Th17 cells, respectively, demonstrates a significant reduction of Th17 cells in SLT, whereas Th1 and Th2 cells are not different between groups (Fig. 3a, b). When CD4 subsets are expressed as a ratio of one cell subtype to another within each subject, there is an increase in the ratio of Th2:Th17 cells in SLT patients, consistent with the

**Table 1 Clinical features of dysregulated immunity in patients with salt-losing tubulopathy (SLT) compared to healthy and disease controls.**

| | Salt-losing tubulopathy, $N = 47$ | Healthy controls, $N = 24$ | Disease controls, $N = 22$ | P-value (SLT vs. healthy controls) | P-value (SLT vs. disease controls) |
|---|---|---|---|---|---|
| **Demographic** | | | | | |
| Age (median; IQR) | 35 (28–43) | 34 (31–45.5) | 42 (22–53) | >0.99 | >0.99 |
| Sex (n, female; %) | 26 (55.3) | 15 (62.5) | 11 (50) | 0.56 | 0.68 |
| **Bacterial Infections** | | | | | |
| Any bacterial infection | 37 (78.7) | 8 (33.3) | 11 (50.0) | **0.0003** | **0.024** |
| Abscess | 10 (21.3) | 2 (8.3) | 0 (0.0) | 0.2 | **0.024** |
| Pneumonia | 8 (17.0) | 3 (12.5) | 2 (9.1) | 0.73 | 0.48 |
| Recurrent upper respiratory tract infection (URTI)[a] | 23 (48.9) | 3 (12.5) | 4 (18.1) | **0.0037** | **0.018** |
| Recurrent urinary tract infection (UTI)[b] | 17 (36.2) | 0 (0.0) | 4 (18.1) | **0.0003** | 0.17 |
| Other severe/recurrent bacterial infection[c,d] | 12 (25.5) | 2 (8.3) | 5 (22.7) | 0.11 | >0.99 |
| **Fungal Infections** | | | | | |
| Any fungal infection | 25 (53.2) | 12 (50.0) | 7 (31.8) | 0.8 | 0.12 |
| Recurrent vaginal (≥2 episodes) or oral candidiasis | 18 (38.3) | 3 (12.5) | 5 (22.7) | **0.029** | 0.27 |
| Fungal fingernail infection | 6 (12.8) | 7 (29.2) | 0 (0.0) | 0.11 | 0.17 |
| Other severe/recurrent fungal infection[e] | 8 (17.0) | 3 (12.5) | 2 (9.1) | 0.74 | 0.48 |
| **Viral Infections** | | | | | |
| Any recurrent viral infection (≥2 episodes) or severe viral infection | 14 (33.3) | 9 (37.5) | 5 (21.7) | 0.79 | 0.57 |
| Recurrent varicella zoster virus (VZV) | 1 (2.3) | 1 (4.2) | 2 (9.1) | >0.99 | 0.27 |
| Recurrent human papilloma virus (HPV) | 9 (21.4) | 3 (12.5) | 2 (9.1) | 0.51 | 0.3 |
| Recurrent Herpes simplex virus (HSV) | 6 (14.2) | 5 (20.8) | 1 (4.5) | 0.51 | 0.41 |
| Other severe/recurrent viral infection[f] | 1 (2.4) | 1 (4.2) | 3 (13.6) | >0.99 | 0.11 |
| **Atopy** | | | | | |
| Any atopic disease | 23 (48.9) | 11 (45.8) | 11 (50.0) | >0.99 | >0.99 |
| Asthma | 8 (17.0) | 5 (20.8) | 4 (18.1) | 0.75 | >0.99 |
| Eczema/dermatitis | 15 (31.9) | 2 (8.3) | 4 (18.1) | **0.039** | 0.26 |
| Hay fever/non-allergic rhinitis | 16 (34.0) | 6 (25.0) | 7 (31.8) | 0.59 | >0.99 |
| **Allergy** | | | | | |
| Any allergic disease | 28 (59.6) | 3 (12.5) | 10 (45.4) | **0.0001** | 0.31 |
| Contact/environmental allergy | 17 (36.2) | 2 (8.3) | 3 (13.6) | **0.012** | 0.08 |
| Food allergy | 8 (17.1) | 0 (0.0) | 3 (13.6) | **0.045** | >0.99 |
| Drug allergy | 12 (25.5) | 1 (4.2) | 6 (27.2) | **0.048** | >0.99 |
| Angiodemea/urticaria unexplained | 5 (10.6) | 0 (0.0) | 0 (0.0) | 0.16 | 0.17 |

Prevalence of each clinical feature is reported as number and percentage. SLT patients are compared to healthy controls and disease controls individually with a two-sided Fisher's exact test (without adjustment for multiple comparison testing).
Source data are provided as a Source data file.
[a]Including tonsillitis, otitis, adenitis and sinusitis; ≥3 in worst year; or tonsillitis requiring tonsillectomy; or URTI requiring admission for intravenous antibiotics.
[b]≥3 in worst year in females; or any UTI in childhood; or any UTI in male.
[c]Requiring antibiotic treatment or admission.
[d]Other severe/recurrent bacterial infection in salt-losing tubulopathy: recurrent dental infection (1); necrotising gingivitis (1); appendicitis/diverticulitis (3); recurrent adenitis (1); skin (impetigo/cellulitis/wound) (3); recurrent gastroenteritis (2); lymphangitis (1); pelvic (1); septicaemia (staph) (1); recurrent mastitis (1).
[e]Other severe/recurrent fungal infection in salt-losing tubulopathy: skin (7); fungal UTI (1).
[f]Other severe/recurrent viral infection in salt-losing tubulopathy: herpetic stomatitis (1).
Significant P values (P ≤ 0.05) are highlighted in bold.

observed findings of increased infection and allergy rate in the SLT cohort (Fig. 3c).

To further investigate IL-17 responses, peripheral blood mononuclear cells (PBMCs) are cultured under optimal Th17 polarising conditions for 7 days. IL-17 expression in CD4+ and CD4− cells, representing Th17 and Tc17 polarisation, respectively, is reduced in SLT patients compared to both healthy and disease controls (Fig. 4a, b, d). Supernatant IL-17 concentrations are reduced in SLT patients compared to HCs (Fig. 4c). Together with the CD4+ cell subset analysis, these data confirm SLT patients have reduced IL-17 responses.

Altered STAT1 and STAT3 phosphorylation are a common cause of inherited defects in IL-17 immunity. We assess STAT1 and STAT3 phosphorylation in SLT lymphocytes, but these are normal demonstrating STAT1 and STAT3 signalling defects are not the cause of defective IL-17 responses in SLT patients (Fig. 5a, b; Supplemental Fig. 4A–C).

Mutations in a number of different ion channels expressed on immune cells lead to immunodeficiency, largely as a result of associated changes in calcium signalling during leucocyte activation[32]. The commonest defective sodium transporter in our SLT cohort was the sodium chloride cotransporter (NCC, *SLC12A3*).

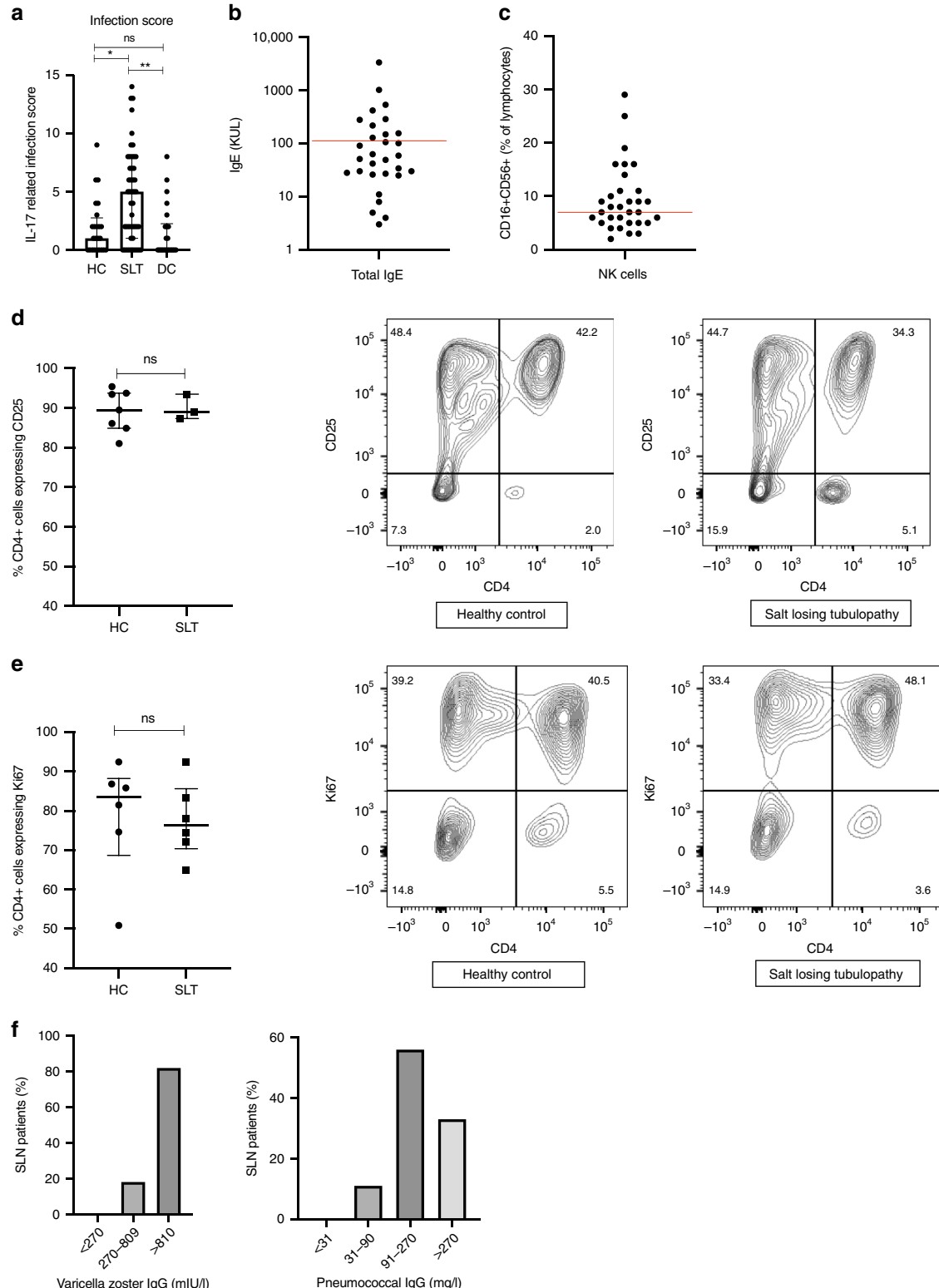

We investigate NCC expression on immune cells and assess whether its inhibition leads to reduced IL-17 responses. NCC is expressed on diverse immune cell subtypes but NCC inhibition with hydrochlorothiazide (HCT; 20 μM) has no effect on T-cell calcium flux or in vitro IL-17 responses in cells from HCs (Fig. 5c and Supplemental Fig. 4D–F). SLT patients have a normal pattern of calcium flux after T-cell activation (Fig. 5d, e). Furthermore, infection score and IL-17 responses are not different according to

the ion transporter affected (Supplemental Fig. 4G). Given these data, reduced IL-17 responses in SLT are not likely to be due to primary perturbations of NCC function or secondary dysregulation of intracellular calcium signalling in immune cells.

**Altered extracellular ions explain SLT immunodeficiency**. We hypothesise that reduced IL-17 responses in SLT are due to their altered in vivo ionic environment of reduced extracellular

**Fig. 2 Initial Immunological analysis of Salt-Losing Tubulopathy patients. a** IL-17-related infection score in SLT patients ($n = 47$), healthy controls ($n = 24$), and disease ($n = 22$) controls: HC 1.0 (0.0–2.8); SLT 5.0 (1.0–8.0); DC 0.0 (0.0–2.3), $p = 0.0006$. Groups are compared using a Kruskal–Wallis test with multiple comparison testing. Error bars represent the interquartile range around the median. **b** Total immunoglobulin E (IgE) levels in SLT patients ($n = 30$). Red line depicts upper limit of normal at 120 KUL; 10/30 (33%) SLT patients had IgE > 120 KUL. **c** NK cells expressed as a proportion of lymphocytes in SLT patients ($n = 32$). Red line depicts lower limit of normal at 7%; 13 (40.6%) SLT patients had NK cells < 7%. **d**. T-cell activation analysis in SLT patients ($n = 3$) and healthy controls ($n = 7$). Proportion of CD4$^+$ cells expressing CD25 after 72 h anti-CD3/anti-CD28 stimulation: HC 89.5% (84.9–95.4), SLT 89.1% (87.3–93.5), $p = 0.99$. Groups are compared with a two-sided Mann–Whitney test. Error bars within the graph represent the interquartile range around the median. Representative FACS contour plots of CD4 and CD25 staining have cell proportions as percentages within each quadrant. **e** T-cell proliferation analysis in SLT patients ($n = 6$) and healthy controls ($n = 6$). Proportion of CD4$^+$ cells expressing Ki67 after 72 h anti-CD3/anti-CD28 stimulation: HC 83.6% (68.7–88.2) and SLT 76.2% (70.4–85.6), $p = 0.48$. Groups are compared with a two-sided Mann–Whitney test. Error bars within the graph represent the interquartile range around the median. Representative FACS contour plots of CD4 and Ki67 staining have cell proportions as percentages within each quadrant. **f**. Varicella zoster and pneumococcal-specific immunoglobulin G levels in SLT patients ($n = 11$). Findings in SLT patients are consistent with the healthy adult population as per assay interpretation data. ns not significant ($p > 0.05$), *$p \leq 0.05$, **$p \leq 0.01$, ***$p \leq 0.001$, ****$p \leq 0.0001$. HC healthy control, SLT salt-losing tubulopathy. Source data are provided as a Source data file.

sodium, potassium, and magnesium. We therefore assess the effect of altering extracellular sodium, potassium, and magnesium concentrations, individually and in combination, on the balance of CD4$^+$ subsets after non-specific T-cell stimulation and on IL-17 responses in control cells cultured under optimal polarising conditions.

We first measure electrolyte concentrations of XVIVO15 media to determine baseline conditions (Supplementary Fig. 5A). We then stimulate PBMCs with anti-CD3 and anti-CD28 for 72 h in different ionic environments and measure the expression of IFNγ, IL-4, and IL-17 in CD4$^+$ cells. The addition of NaCl (0–40 mM), KCl (0–2 mM), and MgCl$_2$ (0–1 mM) results in a dose-dependent reduction in the ratio of expression of IFNγ:IL-17 and IL-4:IL-17 in CD4$^+$ cells (Fig. 6a–c). The addition of each of the ions has no effect on cell viability (Supplementary Fig. 5B), there is no consistent effect on IFNγ:IL-4 expression, and combining multiple ions exacerbates IL-4:IL-17 expression ratio (Fig. 6d). Hence, SLT patients have an extracellular ionic environment, which favours Th2 over Th17 responses, as we had found in our CD4 subset analysis and consistent with their clinical phenotype.

We then assess the effect of altering extracellular ionic concentrations on 7-day IL-17 responses. We demonstrate a dose-dependent increase in Th17 and Tc17 polarisation with the addition of NaCl (0–40 mM), KCl (0–2 mM), and MgCl$_2$ (0–0.8 mM) (Fig. 7a–c). Addition of ions has no effect on cell viability or IFNγ responses (Supplemental Fig. 6A, B). We interrogate the NaCl responses more closely and demonstrate that the polarising effect of NaCl on Th17 and Tc17 cells is consistent in HCs (Fig. 7d, e), that its effect is predominantly mediated by the sodium concentration rather than osmolality (Supplemental Fig. 6C), and that sodium affects both naive and memory T-cell populations (Supplemental Fig. 6D–F). Angiotensin II added to culture conditions increases IL-17 responses whereas aldosterone has no effect (Supplemental Fig. 6G). SLT patients, therefore, have an extracellular ionic environment that reduces IL-17 responses, and this environment may impact IL-17 expression in multiple cell types. Angiotensin II may serve as a compensatory immune mechanism for salt loss.

We then correlate serum biochemical and clinical parameters in SLT with their IL-17-related infection score. Impaired renal function as demonstrated by increased serum creatinine leads to reduced salt loss and blood pressure may act as a surrogate for sodium stores[33,34]. In support of a predominant effect of salt depletion and reduced extracellular sodium on immunity, blood pressure and serum creatinine inversely correlate most closely with infection score (diastolic blood pressure Pearson $r$ −0.37 (95% CI −0.60, −0.09), $p = 0.013$; mean arterial pressure Pearson $r$ −0.32 (95% CI −0.56, −0.03), $p = 0.03$; serum creatinine Pearson $r$ −0.27 (95% CI −0.52, 0.02), $p = 0.06$), whereas there is

no correlation of infection score with serum magnesium or potassium (Supplemental Fig. 7A, B). Moreover, aligned with their differences in renal function, there is a difference in infection score between SLT types grouped according to biochemical phenotype, with infection score highest in patients with GS and EAST syndrome (distal tubule dysfunction and most preserved renal function), and lowest in patients with BS types 1, 2, and 4 (loop dysfunction and most impaired renal function) (Fig. 7f, g).

**SLT IL-17 responses can be rescued in vitro by salt**. The intracellular pathway by which sodium increases Th17 polarisation includes the upregulation of SGK1, NFAT5, and the IL23-receptor (IL-23R)[12,13]. We assess whether this pathway is intact in SLT patients. We demonstrate that SLT lymphocyte expression of SGK1 and NFAT5 are not reduced compared to HCs, neither is IL-23R expression on CD4$^+$ cells (Fig. 8a–e); IL23-R expression is in fact increased in non-CD4 lymphocytes in SLT patients (Fig. 8B and Supplemental Fig. 8A).

Given the preservation of this pathway, we hypothesised that in vitro IL-17 responses can be rescued in SLT patients by supplementing culture conditions with salt. Indeed, the addition of 40 mM NaCl to culture conditions upregulates IL-17 responses in SLT such that Th17 and Tc17 polarisation are equivalent to healthy and disease control polarisation under standard conditions (Fig. 8f, g). Salt responsiveness of in vitro IL-17 responses in SLT patients is no different to controls, and there is no difference in salt responsiveness according to SLT type (Supplemental Fig. 8B–D). These data demonstrate that salt supplementation ameliorates IL-17 response defects in SLT in vitro, but whether this is feasible or effective clinically in vivo is unknown.

**Innate immune responses in SLT patients**. Polarisation of Th17 cells requires antigen presentation by antigen-presenting cells in the presence of Th17 polarising cytokines produced by innate immune cells. Defective IL-17 responses can, therefore, be a consequence of impaired innate immunity. We assess innate immune responses in SLT patients by measuring cytokine production in response to a range of stimuli in whole blood assays. We demonstrate no reduction in innate responses; indeed, production of several innate cytokines (e.g. IL-1, IL-6, and TNF) is increased in SLT patients compared to controls (Table 2).

We subsequently analyse monocyte responses in more detail in SLT patients. We find no difference in monocyte subsets (classical, intermediate, and non-classical) or in monocyte expression of TNF in SLT patients compared to controls in the experiment we use (Supplemental Fig. 9A–C). Moreover, adding 40 mM NaCl, 2 mM KCl, 1 mM MgCl$_2$, or aldosterone

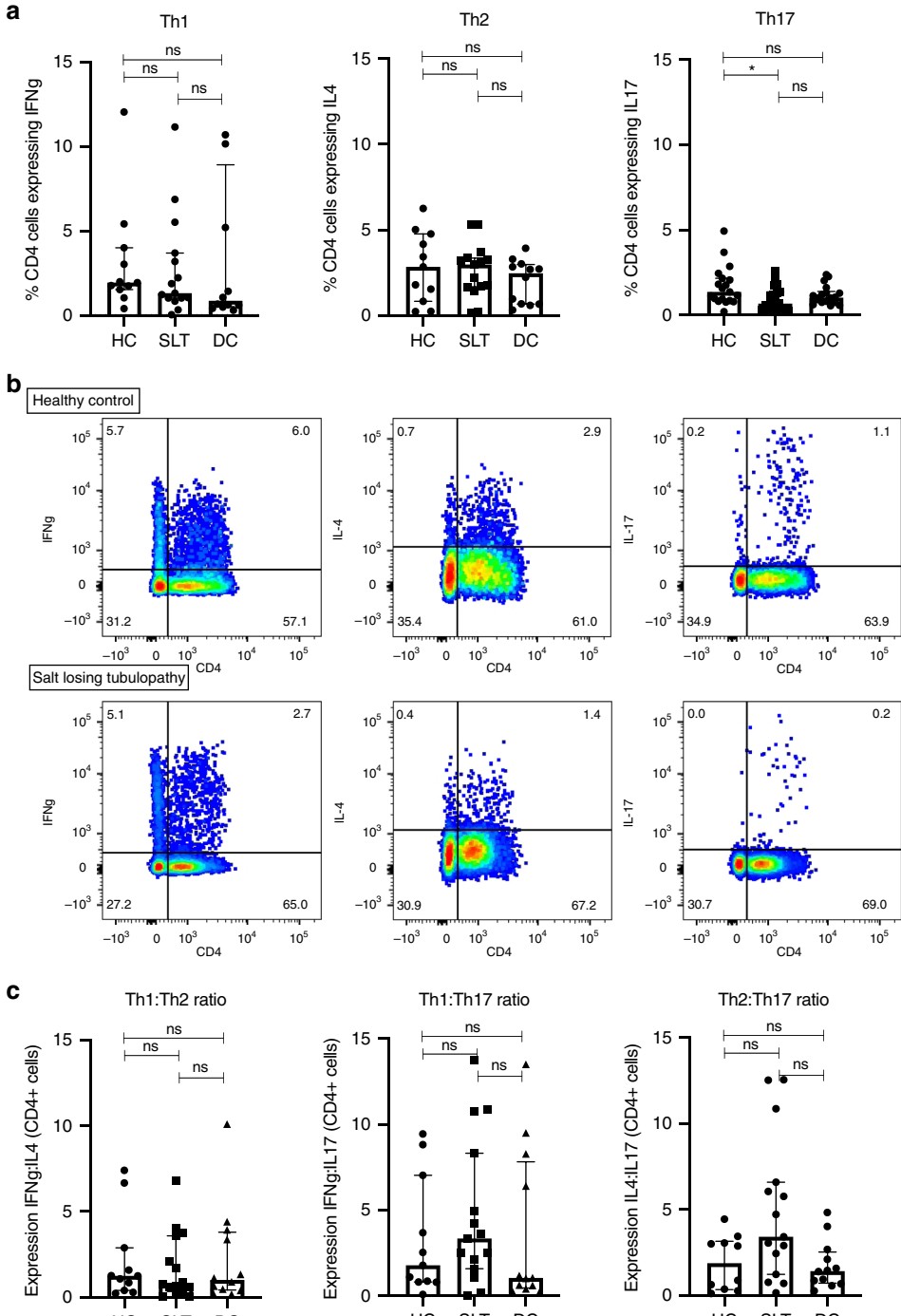

**Fig. 3 CD4 subset analysis in salt-losing tubulopathy patients and controls. a** IFNγ, IL-4, and IL-17 expression in CD4+ cells after 4 h stimulation with phorbol myristate acetate and ionomycin in SLT patients (n = 15), healthy controls (n = 11), and disease controls (n = 12). Cytokine expression is reported as percentage of CD4+ cells. Groups are compared with a Kruskal–Wallis test with Dunn's multiple comparison testing (shown with significance bars). Error bars represent interquartile range around the median. IFNγ expression: HC 1.9% (1.5–4.0), SLT 1.3% (1.0–3.7), DC 0.9% (0.5–8.9), p = 0.53. IL-4 expression: HC 2.8% (0.8–4.8), SLT 3.0% (1.7–3.3), DC 2.5% (0.7–3.0), p = 0.52. IL-17 expression: HC 1.4% (0.9–2.2), SLT 0.7% (0.3–1.4), DC 1.0% (0.8–1.4), p = 0.038. **b** Representative FACS dot plots of IFNγ, IL-4, and IL-17 expression in CD4+ cells in a SLT patient and HC. Cell proportions as percentages are documented within each quadrant. **c** Ratio of IFNγ, IL-4, and IL-17 expression in each subject in SLT patients (n = 15), HC (n = 11), and DC (n = 12). Groups are compared with a Kruskal–Wallis test with Dunn's multiple comparison testing (shown with significance bars). Error bars represent interquartile range around the median. IFNγ:IL-4 (Th1:Th2): HC 1.2 (0.4–2.9), SLT 0.7 (0.3–3.9), DC 1.0 (0.4–3.9), p = 0.82. IFNγ:IL-17 (Th1:Th17): HC 1.8 (0.8–7.0), SLT 3.3 (1.6–8.3), DC 1.0 (0.5–7.8), p = 0.40. IL-4:IL-17 (Th2:Th17): HC 1.9 (0.4–3.2), SLT 3.4 (1.2–6.6), DC 1.4 (0.7–2.5), p = 0.05. ns not significant (p > 0.05), *p ≤ 0.05, **p ≤ 0.01, ***p ≤ 0.001, ****p ≤ 0.0001, HC healthy control, SLT salt-losing tubulopathy, DC disease control. Source data are provided as a Source data file.

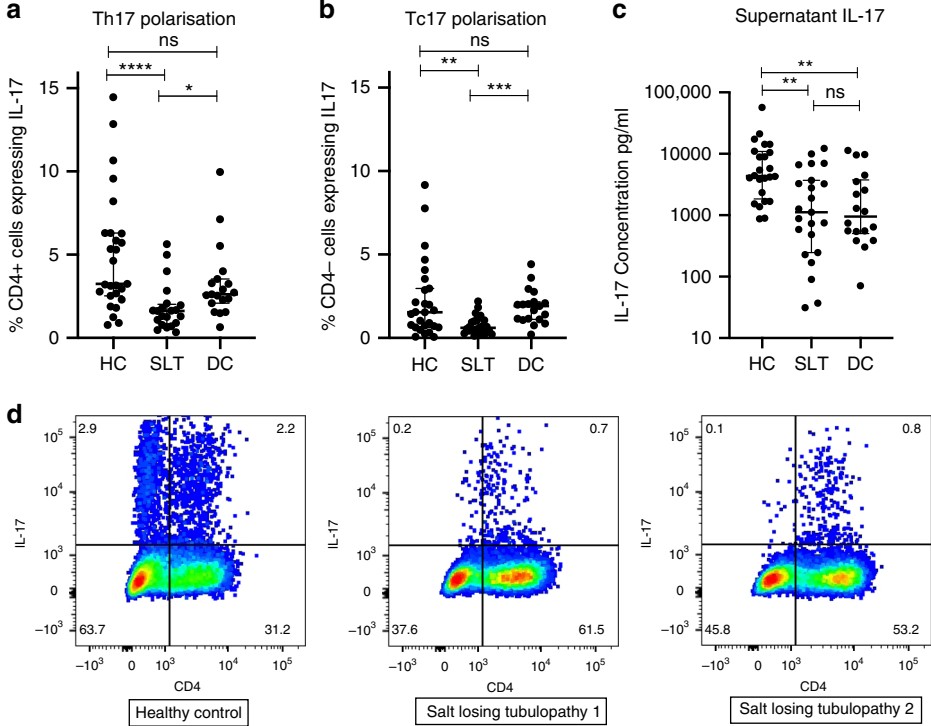

**Fig. 4 IL-17 responses in salt-losing tubulopathy patients and controls. a** IL-17 expression in CD4$^+$ cells (Th17) after 1-week culture of HC ($n = 27$), SLT ($n = 21$), and DC ($n = 19$) PBMCs stimulated with anti-CD3, anti-CD28, IL-1, IL-6, IL-21, IL23, and TGFβ: HC 3.2% (2.5–6.3), SLT 1.6% (0.8–2.0), DC 2.6% (2.1–3.5), $p = 0.0001$. Groups are compared with a Kruskal–Wallis test with Dunn's multiple comparison testing (shown with significance bars). Error bars represent interquartile range around the median. **b** IL-17 expression in CD4− cells (Tc17) after 1-week culture of HC ($n = 27$), SLT ($n = 25$), and DC ($n = 19$) PBMCs stimulated with anti-CD3, anti-CD28, IL-1, IL-6, IL-21, IL23, and TGFβ: HC 1.5% (0.6–3.0), SLT 0.6% (0.3–1.1), DC 1.9% (1.1–2.2), $p = 0.0005$. Groups are compared with a Kruskal–Wallis test with Dunn's multiple comparison testing (shown with significance bars). Error bars represent interquartile range around the median. **c** Supernatant IL-17 concentrations cells after 1-week culture of HC ($n = 24$), SLT ($n = 23$), and DC ($n = 18$) PBMCs stimulated with anti-CD3, anti-CD28, IL-1, IL-6, IL-21, IL23, and TGFβ: HC 4287 pg/ml (1681–10,487), SLT 1117 pg/ml (249–3709), DC 951 pg/ml (503–3768), $p = 0.0012$. Groups are compared with a Kruskal–Wallis test with Dunn's multiple comparison testing (shown with significance bars). Error bars represent interquartile range around the median. **d** Representative FACS dot plots of IL-17 expression in CD4$^+$ cells in a HC and 2 SLT patients. Cell proportions as percentages are documented within each quadrant. ns not significant ($p > 0.05$), *$p \leq 0.05$, **$p \leq 0.01$, ***$p \leq 0.001$, ****$p \leq 0.0001$. SLT salt-losing tubulopathy, HC healthy control, DC disease control, PBMCs peripheral blood mononuclear cells. Source data are provided as a Source data file.

(10–100 nM) to stimulating conditions has no effect on monocyte TNF expression (Supplemental Fig. 9D). Reduced innate immune responses are therefore not the cause of reduced IL-17 responses and increased infection in SLT patients.

We finally conduct an additional analysis of NK cells in SLT to further characterise NK cell subsets and assess NK cell function. In this analysis, CD45$^+$CD3$^-$CD56$^+$CD16$^+$ (NK) cells account for 10.2% (5.9–13.8) of lymphocytes in HCs and 6.9% (3.9–10.0) in SLT patients ($p = 0.1$, Supplemental Fig. 10A). 8 (50%) SLT patients have NK cells of less than 7% of lymphocytes, but so do 5 (29%) controls ($p = 0.3$). The proportion of CD56$^+$ cells, however, expressing CD16 (the Fc receptor) is reduced in SLT patients (Supplemental Fig. 10B, D). To assess NK cell function, PBMCs are stimulated with IL-12 and IL-18 for 4 h and NK cell expression of IFNγ determined. IFNγ expression is not different between SLT and controls (Supplemental Fig. 10C).

Although the effect of extracellular sodium has been assessed in large parts of the immune system, its effect on NK cells has not. We, therefore, stimulate NK cells in the presence of additional NaCl (+40 mM). Adding NaCl to stimulating conditions increases the proportion of lymphocytes expressing CD56, but IFNγ expression by CD56$^+$ cells is reduced (Supplemental Fig. 10E, F). There is therefore a trend to a reduction in NK cell number in SLT patients with reduced expression of the Fc receptor on CD56$^+$ cells, demonstrating NK cells in SLT patients have a tolerant phenotype.

## Discussion

Studies to date on the effect of altered sodium balance on immunity have largely focused on salt loading and how this impacts inflammatory or autoimmune disease. Alongside the inflammatory effect of sodium demonstrated on multiple immune cells in vitro and in animal models, increased sodium intake in patients has been linked to the development of rheumatoid arthritis, correlates with disease severity in multiple sclerosis, and worsens renal allograft outcomes independently of any blood pressure effect[35–37]. These data imply that reducing salt intake also impacts immunity and may ameliorate diverse inflammatory states, but this hypothesis is untested and the immunological consequences of chronic salt depletion are unknown. Moreover, the evolutionary pressures linking immune function to sodium are unclear. Long-term salt restriction studies are logistically challenging and difficult to adhere to; therefore to understand the consequences of reduced sodium balance, we investigated immunity in patients with rare inherited SLTs. These patients provide a unique in vivo model of chronic salt depletion and an assessment of their immunity was previously unexplored.

Novel findings from long-term sodium balance studies have demonstrated a third compartment of sodium stored at interstitial sites in the muscle and skin. It is proposed that the skin contains an osmotic gradient-generating system akin to the countercurrent mechanism in the kidney[38]. At both sites increased sodium concentrations prevent excess water loss but

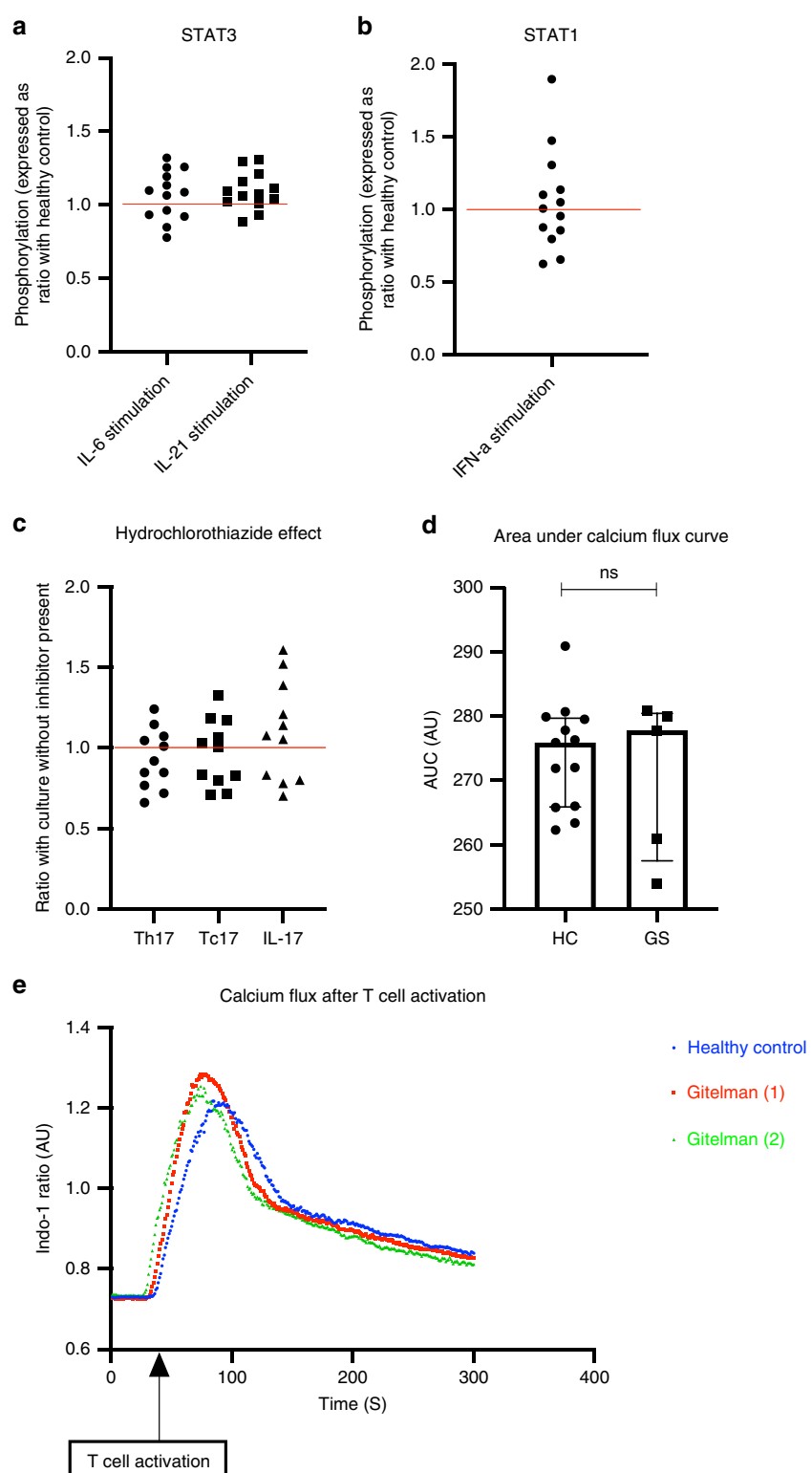

may also provide protection from infection[11,18]. Although the sodium transport mechanisms in the kidney are well described, ion channels and transporters involved in skin sodium transport are largely unknown. Visualisation and measurement of skin sodium stores with magnetic resonance imaging ($^{23}$Na-MRI) has demonstrated that sodium concentrations at these sites increase with age and in disease states that result in sodium loading such as heart failure and chronic kidney disease[33,34,39–41]. In SLT patients, adaptive mechanisms such as increased salt appetite and

enhanced reabsorption of sodium in unaffected nephron segments are upregulated to prevent catastrophic sodium loss[42]. Although SLT patients waste sodium, how this affects sodium stores or if this impacts on total body sodium balance in the long term is unexplored. Moreover, if the defective ion transport mechanisms in SLT patients directly impact keratinocyte sodium transport is unknown. Using $^{23}$Na-MRI, we demonstrated that sodium stores are reduced in SLT patients compared to controls. This reduction was seen in SLT patients with defects in both renal

**Fig. 5 Initial investigation of impaired IL-17 responses in salt-losing tubulopathy patients. a** Phosphorylation of STAT3 in CD4$^+$ cells after stimulation with IL-6 and IL-21 in SLT patients ($n = 13$). Phosphorylation is expressed as ratio of upregulation of pSTAT3 in SLT compared to the HC assessed in the same experiment ($n = 4$ HCs assessed across 4 experiments). Red line drawn at ratio of 1 representing no difference between SLT and HC. **b** Phosphorylation of STAT1 in CD4$^+$ cells after stimulation with IFNα in SLT patients ($n = 13$). Phosphorylation is expressed as ratio of upregulation of pSTAT1 in SLT compared to the HC assessed in the same experiment ($n = 4$ HCs assessed across 4 experiments). Red line drawn at a ratio of 1 representing no difference between SLT and HC. **c** Th17 and Tc17 polarisation, and supernatant IL-17 concentration in cell culture experiments (11 experiments in 11 healthy controls) with the addition of hydrochlorothiazide (HCT) 20 μM to culture conditions. Each variable is expressed as a ratio to the readout in standard culture conditions; red line drawn at ratio of 1 representing no difference in readouts with the addition of HCT. **d** Area under the calcium flux curve in healthy controls ($n = 13$) and Gitelman syndrome patients ($n = 5$) (as determined using the ratiometric calcium dye Indo-1 gated on CD4$^+$ cells) after T-cell activation with anti-CD3: HC 275.9 AU (265.9–279.9); and Gitelman Syndrome (GS) 277.8 AU (257.5–280.5), $p = 0.94$. Compared with a two-sided Mann–Whitney test. Error bars represent interquartile range around the median. **e** Representative calcium flux curves in CD4$^+$ cells after T-cell activation in a healthy control and two patients with GS. ns not significant ($p > 0.05$), *$p \leq 0.05$, **$p \leq 0.01$, ***$p \leq 0.001$, ****$p \leq 0.0001$, SLT salt-losing tubulopathy, HC healthy control, DC disease control, GS Gitelman syndrome. Source data are provided as a Source data file.

specific (BS1) and systemic (GS) transporters suggesting the predominant reason for reduced skin sodium is renal sodium wasting as opposed to a primary defect in sodium transport in the skin. This alteration in interstitial sodium combined with reduced serum potassium and magnesium concentrations mean immune cells in SLT patients are exposed to a unique combination of altered extracellular ionic cues, which we propose impacts immune function.

The presentation of SLT patients may be through the identification of biochemical abnormalities incidentally on blood tests performed for other reasons, or with an array of relatively non-specific symptoms. These include cramps, weakness, dizziness, and fatigue, and such symptoms have been traditionally attributed directly to electrolyte derangement or relative hypovolaemia. However, there are no consistent data demonstrating a correlation between symptom burden and electrolyte abnormalities, and symptoms are often persistent and disabling even when electrolytes are replete. Moreover, some recognised clinical features, such as febrile episodes in childhood in GS, theoretically cannot be explained by a direct electrolyte effect[27,43–45]. Our data demonstrating immunodeficiency in SLT provide an explanation for these symptoms, and we describe novel immune-mediated features that have not been appreciated in SLT patients before. In our cohort, the persistent nature of bacterial and fungal infections alongside recurrent allergic episodes carried significant morbidity. Moreover, an association between SLT and autoimmunity (e.g. autoimmune thyroid disease, Sjögren's syndrome, autoimmune hepatitis, and vasculitis) is recognised[22–24,26,28]; findings in our cohort confirm this association and our demonstration of dysregulated immunity provide an explanation for this link.

Th17 cells are pro-inflammatory, and provide protection against extracellular bacterial and fungal infections, particularly at mucosal surfaces[46]. Polarisation requires stimulation by IL-1β, IL-6, and TGF-β, whereas full maturation requires IL-21 and IL-23. Th17 cells secrete various cytokines including IL-17, which induces release of other pro-inflammatory cytokines and chemokines, and is important for neutrophil recruitment. Defective IL-17 responses may be inherited and result in CMC syndrome, or may be acquired in patients with thymoma[47] (production of neutralising antibodies to IL-17) or in patients taking anti-IL-17 monoclonal antibodies for the treatment of autoimmune disease[48]. We demonstrate that the primary cause for immunodeficiency in SLT patients is reduced IL-17 responses and the mucosal nature of bacterial and fungal infections in SLT is consistent with the phenotype of patients with inherited IL-17 defects (Supplemental Table 6). Syndromic CMC is most commonly due to STAT1 phosphorylation abnormalities[6], but these were absent in SLT patients, as were alterations in STAT3-mediated cytokine signalling[31,49]. We describe, therefore, a novel cause of IL-17 immunodeficiency, through an ionic mechanism not previously reported before.

Immunodeficiency as a result of altered ionic flux in immune cells is largely described in humans from inherited ion channe-lopathies that result in impaired store-operated calcium entry (SOCE) during T-cell activation[50]. This may be due to calcium release-activated calcium current (CRAC) channel mutations leading to impaired calcium release from the endoplasmic reticulum (e.g. stromal interaction molecule 1 [STIM1] mutations) or impaired extracellular calcium entry across the plasma membrane (e.g. calcium release-activated calcium modulator 1 [ORAI1] mutations)[51–56]. Impaired SOCE also can result from a magnesium channel defect due to pathogenic mutations in Magnesium Transporter 1 (MAGT1), which results in X-linked immunodeficiency with magnesium defect, EBV viraemia and neoplasia (XMEN) syndrome[57,58]. Moreover, in mice, impaired SOCE due to deletion of ORAI1 or STIM1 impairs Th17 polarisation[59,60], and sodium has been shown to exert its pro-inflammatory effect in dendritic cells through epithelial sodium channel (ENaC) flux altering calcium signalling[61]. In SLT, we demonstrate that calcium flux is unaffected after T-cell activation and patients with inactivating ENaC mutations (pseudohypoaldosteronism type 1) were not included in the SLT cohort. Although we have demonstrated the presence of other sodium transport mechanisms on diverse immune cells, this unaltered calcium flux, alongside the lack of an effect of sodium chloride cotransporter inhibition on Th17 polarisation, and the inclusion of patients with multiple ion channel/transporter mutations make reduced function of channel/transporter activity on immune cells an unlikely reason for altered immunity.

SLT patients are unique in their characteristic extracellular ionic environment, directly and indirectly related to renal salt wasting. In the light of recent evidence highlighting the effect of sodium on IL-17 responses[12,13,16,17,62,63], and on previous evidence demonstrating an effect of other extracellular ions (e.g. magnesium) on immunity[64], we hypothesised that reduced IL-17 responses are due to the altered ionic environment in vivo in SLT patients. We demonstrated that the addition of sodium, potassium and magnesium, within a clinically relevant range, promoted both a reduction in the ratio of Th2:Th17 polarisation during 3-day T-cell stimulation experiments, and an increase in Th17 and Tc17 polarisation during 7-day experiments in culture conditions optimal for IL-17 production. These data show that the extracellular ionic environment typically found in SLT patients is inhibitory to protective IL-17 responses. Our experiments confirm previous data demonstrating sodium promotes IL-17 responses[12,13,16,17,62,63], but also highlight a potential role of sodium impacting Th2 and NK responses, and suggest an influence of other cations on Th17 polarisation. We also provide evidence for the first time that sodium depletion affects immunity and leads to IL-17-mediated immunodeficiency. Moreover, within the SLT cohort, blood pressure and serum creatinine

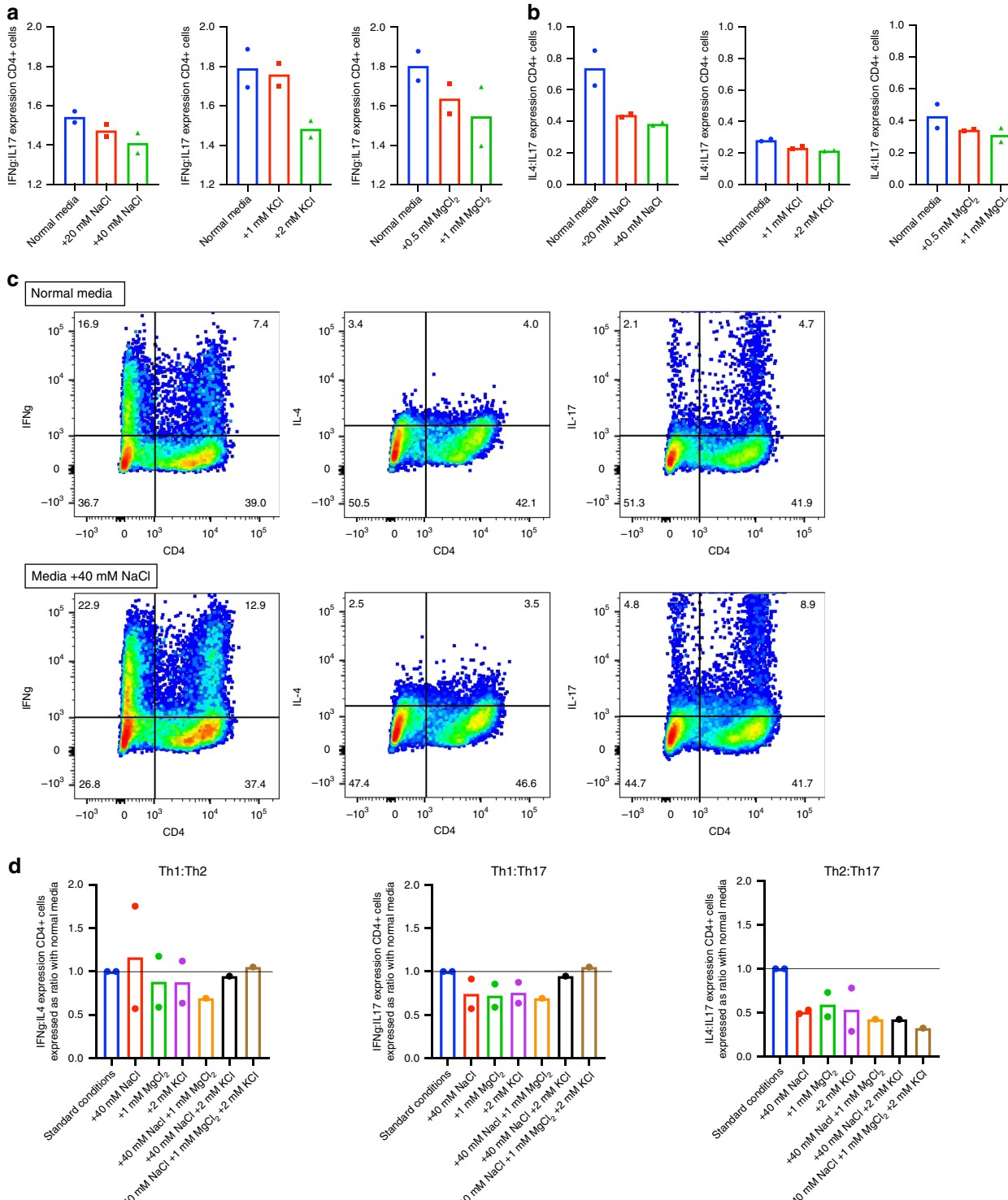

**Fig. 6 Effect of altering extracellular ion concentration within a physiological range on CD4 subset balance. a** Effect of NaCl (0–40 mM), KCl (0–2 mM), and MgCl$_2$ (0–0.8 mM) during T-cell activation with anti-CD3/anti-CD28 on the ratio of expression of IFNγ:IL-17 in CD4$^+$ cells, representative of Th1:Th17 cells. Data shown represent dose–response in a healthy control. Mean of technical duplicates is plotted. **b** Effect of NaCl (0–40 mM), KCl (0–2 mM), and MgCl$_2$ (0–0.8 mM) during T-cell activation with anti-CD3/anti-CD28 on the ratio of expression of IL-4:IL-17 in CD4$^+$ cells, representative of Th2:Th17 cells. Data shown represent dose–response in a healthy control. Mean of technical duplicates is plotted. **c** Representative FACS dot plots of the effect of +40 mM NaCl on the expression of IFNγ, IL-4, and IL-17 in CD4$^+$ cells from a healthy control. Cell proportions as percentages are documented within each quadrant. **d** CD4$^+$ subset balance after T-cell activation with anti-CD3/anti-CD28 in the presence of NaCl (40 mM), KCl (2 mM), and MgCl$_2$ (1 mM) used alone and in combination. Data show subset balance reported as a ratio to standard conditions in healthy controls ($n = 2$). The median ratio is plotted. Lines drawn at a ratio of 1 represents no difference in subset balance in the presence of ions compared to standard conditions. Source data are provided as a Source data file.

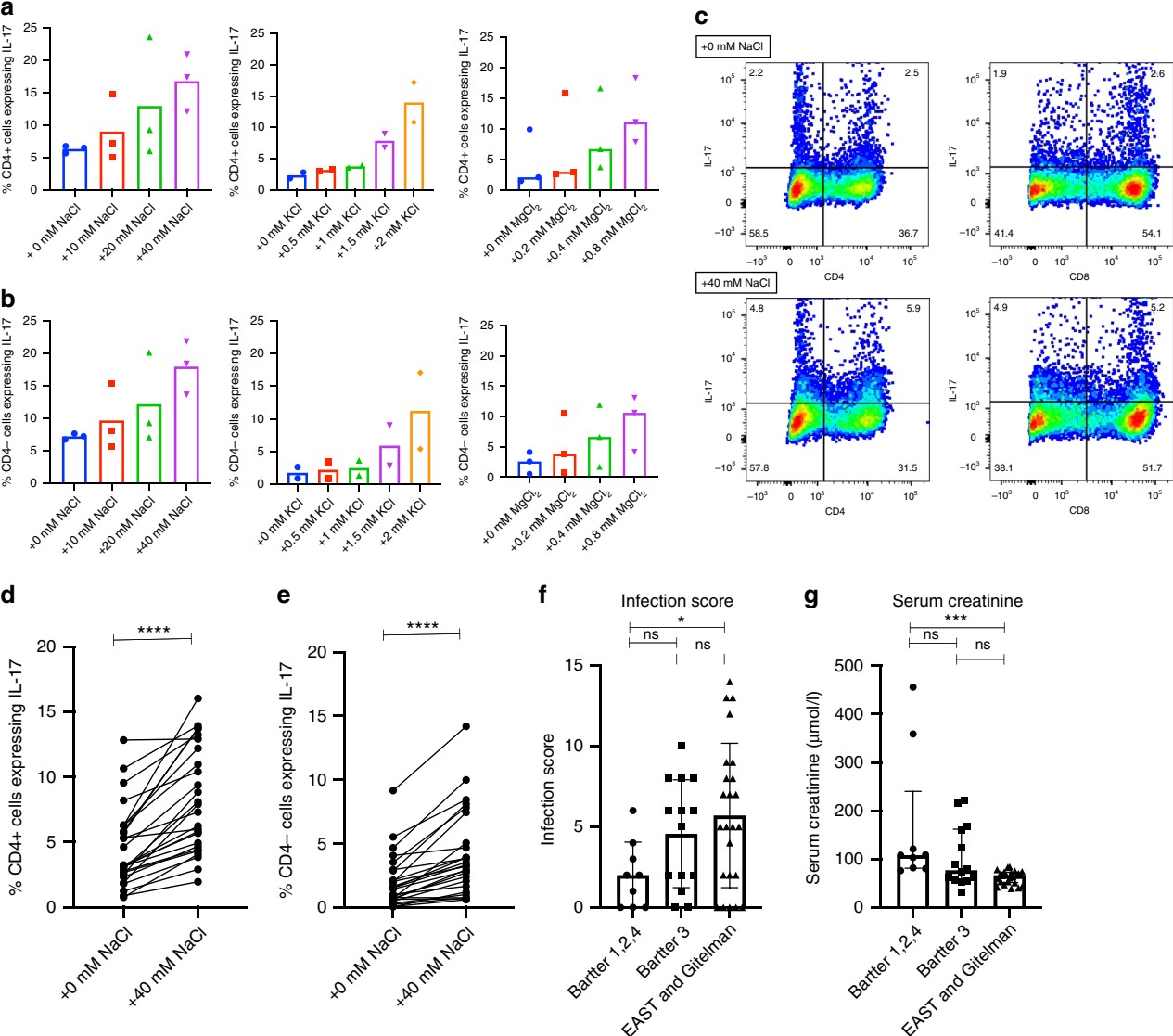

**Fig. 7 Effect of altered extracellular ion concentration on Th17 and Tc17 polarisation. a** Effect of NaCl (0–40 mM; $n = 1$ healthy control), KCl (0–2 mM; $n = 2$ healthy controls), and MgCl$_2$ (0–0.8 mM; $n = 3$ healthy controls) on Th17 polarisation after PBMC activation under optimal Th17 polarising conditions. Mean of technical triplicates is plotted for NaCl effect. In other graphs the median for controls is plotted and data points represent mean of technical replicates in each participant. **b** Effect of NaCl (0–40 mM; $n = 1$ healthy control), KCl (0–2 mM; $n = 2$ healthy controls), and MgCl$_2$ (0–0.8 mM; $n = 3$ healthy controls) on Tc17 polarisation after PBMC activation under optimal Th17 polarising conditions. Mean of technical triplicates is plotted for NaCl effect. In other graphs the median for controls is plotted and data points represent mean of technical replicates in each participant. **c** Representative FACS dot plots of IL-17 expression in CD4$^+$ (Th17) and CD8$^+$ (Tc17) healthy control cells after activation in normal media and in media supplemented with 40 mM NaCl. Cell proportions as percentages are documented within each quadrant. **d**. Th17 polarisation of healthy control ($n = 26$) cells cultured in normal media and in media supplemented with 40 mM NaCl. Th17 cells (as a percentage of CD4$^+$ cells): normal media 3.2% (2.5–6.3); media +40 mM NaCl 7.6% (4.6–12.4); $p < 0.0001$. Conditions are compared with a two-sided Wilcoxon test. Data points and connecting lines represent change in polarisation within each healthy control. **e** Tc17 polarisation of healthy control ($n = 26$) cells cultured in normal media and in media supplemented with 40 mM NaCl. Tc17 cells (as a percentage of CD4$^-$ cells): normal media 1.5% (0.6–2.9); media +40 mM NaCl 3.3% (1.5–5.6); $p < 0.0001$. Conditions are compared with a two-sided Wilcoxon test. Data points and connecting lines represent change in polarisation within each healthy control. **f** IL-17-related infection score according to SLT biochemical phenotype: BS types 1, 2, and 4 [$n = 9$] = 2.0 ± 2.1 points; BS type 3 [$n = 14$] = 4.6 ± 3.3 points; GS and EAST [$n = 24$] = 5.7 ± 4.5 points; $p = 0.05$. Groups are compared using a one-way analysis of variance with multiple comparison testing (shown with significance bars). Error bars represent standard deviation around the mean. **g** Serum creatinine according to SLT biochemical phenotype: BS types 1, 2, and 4 [$n = 9$] = 108 µmol/l (81–240), BS type 3 [$n = 14$] = 77 µmol/l (56–162), GS and EAST [$n = 24$] = 66.5 µmol/l (52–73); $p = 0.0005$. Groups are compared with a Kruskal–Wallis test and multiple comparison testing (shown with significance bars). Error bars represent interquartile range around the median. ns not significant ($p > 0.05$), *$p \leq 0.05$, **$p \leq 0.01$, ***$p \leq 0.001$, ****$p \leq 0.0001$. Source data are provided as a Source data file.

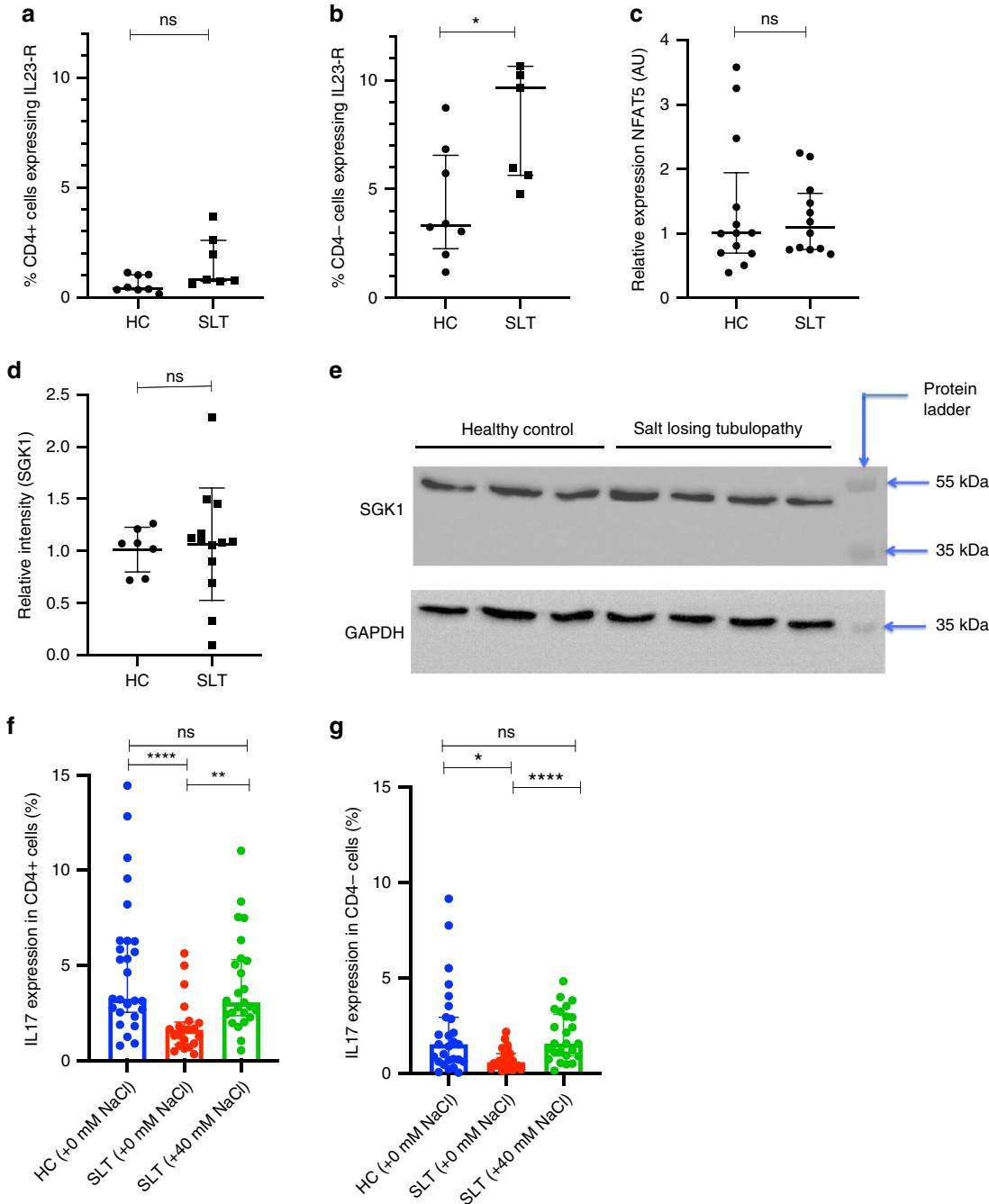

inversely correlated to infection score, and hence we propose that extracellular sodium as opposed to alterations in other ions is the main determinant of infection risk. This requires confirmation with direct measurement of sodium stores in a larger number of SLT patients as part of future study.

Given this proposal of ionic changes altering immunity in SLT, we lastly tested whether manipulation of the extracellular ionic environment in vitro could be used to augment SLT IL-17 responses, as we had shown in controls. We demonstrated that the intracellular pathway that mediates sodium-driven Th17 polarisation is intact in SLT patients and that Th17 polarisation could be rescued with the addition of NaCl to culture conditions. This raises the possibility that better correction of extracellular ionic concentrations in vivo in SLT patients may improve IL-17 responses and mitigate infection risk and warrants further study.

Moreover, our data provide the basis for investigating whether manipulation of the extracellular ionic environment may be used as a therapeutic strategy in IL-17-mediated inflammatory diseases.

We demonstrated that SLT patients do not have alterations in naive and memory cell populations, and that the extracellular ionic environment impacts memory as well as naive cells confirming recent reports[65]. Hence, we used PBMCs as opposed to isolated cells for IL-17 polarisation experiments. We did not assess all CD4$^+$ subtypes and any effect on regulatory T cells or T follicular helper cells, both of which may be affected by sodium[14,15,66], in SLT patients was unexplored. Similarly, sodium has been shown to impact dendritic cell function[61], but the clinical phenotype of SLT patients (absent recurrent viral and mycobacterial infections) and the specific defect in IL-17 responses were inconsistent with primary defects in dendritic

**Fig. 8 Expression of the components of the intracellular pathway that mediate salt-driven IL-17 inflammation and salt rescue of in vitro IL-17 responses in salt-losing tubulopathy patients. a** IL-23 receptor expression on unstimulated CD4$^+$ cells in healthy controls ($n = 8$) and SLT patients ($n = 7$): HC 0.42% (0.34–1.01), SLT 0.79% (0.73–2.60), $p = 0.07$. Groups are compared with a two-sided Mann–Whitney test. Error bars represent interquartile range around the median. **b** IL-23 receptor expression on unstimulated CD4$^-$ cells in healthy controls ($n = 8$) and SLT patients ($n = 7$): HC 3.34% (2.26–6.56), SLT 9.67% (5.64–10.65), $p = 0.02$. Groups are compared with a two-sided Mann–Whitney test. Error bars represent interquartile range around the median. **c** Relative lymphocyte NFAT5 expression in healthy controls ($n = 13$) and SLT patients ($n = 11$): HC 1.01 AU (0.70–1.95), SLT 1.10 AU (0.76–1.62), $p = 0.85$. Groups are compared with a two-sided Mann–Whitney test. Error bars represent interquartile range around the median. **d** Relative lymphocyte SGK1 expression in healthy controls ($n = 7$) and SLT patients ($n = 13$): relative intensity HC = 1.07 AU (0.73-1.2), SLT = 1.09 AU (0.80–1.31), $p = 0.70$. Groups are compared with a two-sided Mann–Whitney test. Error bars represent interquartile range around the median. **e** Representative western blot of SGK1 expression in HCs and SLT patients. SGK1 expression was determined in SLT patients ($n = 13$) and healthy controls ($n = 7$) across three experiments. **f** Th17 polarisation in HC ($n = 27$) and SLT patients ($n = 25$) in standard media, and in SLT patients in media supplemented with 40 mM NaCl: HC + 0 mM NaCl 3.2% (2.5–6.3); SLT + 0 mM NaCl 1.6% (0.8–2.0); SLT + 40 mM NaCl 3.1% (2.4–5.3); $p = <0.0001$. Groups are compared using a Kruskal–Wallis test with Dunn's multiple comparison testing (shown with significance bars). Error bars represent interquartile range around the median. **g** Tc17 polarisation in HC ($n = 27$) and SLT patients ($n = 25$) in standard media, and in SLT patients in media supplemented with 40 mM NaCl: HC + 0 mM NaCl 1.5% (0.6–3.0); SLT + 0 mM NaCl 0.6% (0.3–1.1); SLT + 40 mM NaCl 1.6% (0.9–3.1); $p = 0.0007$; groups are compared using a Kruskal–Wallis test with Dunn's multiple comparison testing (shown with significance bars). Error bars represent interquartile range around the median. ns not significant ($p > 0.05$), *$p \leq 0.05$, **$p \leq 0.01$, ***$p \leq 0.001$, ****$p \leq 0.0001$. HC healthy control, SLT salt-losing tubulopathy, AU arbitrary units. Source data are provided as a Source data file.

cells[67,68]. Moreover, innate cytokine responses in whole blood assays were unaffected and hence further functional assessment of dendritic cells was not undertaken. We measured calcium flux during T-cell activation but were unable to measure sodium flux in patients or controls and hence were unable to confirm that SLT patients do not have a primary defect in ion transport mechanisms on immune cells. SLT patients have an activated renin–angiotensin–aldosterone system, albeit largely as a consequence of salt wasting, and while we assessed the effect of aldosterone and angiotensin II on Th17 polarisation, we did not assess the direct effect of renin, which has been shown to have an independent immunomodulatory role[69]. We also did not assess the impact of altered prostaglandin E2, which is known to affect immunity[70] and which is increased in some patients with BS types 1, 2, and 4. Finally, we did not investigate the in vivo location of the effect of sodium and other ions on immunity, which remains an important unknown from this and other studies on the effect of salt on immunity.

In conclusion, we describe a novel immunodeficiency in SLT patients, who have increased mucosal bacterial and fungal infections, alongside other clinical features of dysregulated immunity including autoimmune phenomena, dermatitis, and allergic disease. These clinical features are associated with impaired IL-17 responses, which we propose are due to sodium wasting and the unique extracellular ionic environment found in SLT patients. We highlight the role of multiple extracellular ions on immune cell activation states and demonstrate that SLT IL-17 responses may be rescued in vitro by salt. The manipulation of the extracellular ionic environment may, therefore, be of therapeutic benefit in SLT patients to mitigate infection risk.

## Methods

**SLT and control cohorts.** Patients with genotyped BS, GS, or EAST syndrome were recruited from the Royal Free Hospital and Great Ormond Street Hospital Tubular Disorders Clinics, London, UK. Demographic, biochemical, and clinical data on the day of recruitment were recorded. Biochemical parameters were compared between patients with loop dysfunction (BS types 1, 2, and 4), distal tubule dysfunction (GS and EAST syndrome), and a mixed phenotype (BS type 3).

SLT patients underwent a structured clinical history focused on clinical features of dysregulated immunity. This was also undertaken in age and sex-matched healthy ($n = 24$) and disease ($n = 22$) controls. Disease controls included patients attending the Tubular Disorders Clinics, but who had no evidence of a significant salt-wasting phenotype. These patients were recruited from the same clinics as the SLT patients, and they underwent the same structured clinical history and biochemical analysis. Conditions included: proximal Tubulopathy = 7; monogenic hypertension = 5 (familial hyperkalaemic hypertension = 3; STX16 mutation = 2); distal renal tubular acidosis/nephrocalcinosis = 3; isolated renal magnesium wasting = 1; hyperparathyroidism = 1; tubulointerstitial nephritis = 2; medullary

sponge kidney = 1; diabetes insipidus = 1; non renal hypokalaemia = 1 (Supplementary Table 2). Healthy controls were volunteers who worked within the hospital or university and who had no known medical conditions. The prevalence of each clinical variable was compared between SLT and control groups.

An infection score was subsequently determined for each subject related to infections known to arise from defects in IL-17-mediated immunity. To do this we used the infection-related scoring system in the diagnostic criteria of Hyper-IgE syndrome, which results from inherited defects in IL-17 immunity (Supplemental Table 5). IL-17-related infection scores were compared between SLT and controls. We also compared infection scores between SLT subtypes (according to defective transporter and biochemical group) and we correlated infection score to serum biochemical and clinical parameters in SLT patients.

**$^{23}$Na-MRI imaging.** To determine skin and muscle sodium storage, SLT patients ($n = 6$) and HCs ($n = 4$) underwent $^{23}$Na-MRI imaging of the lower right limb. Tubes of known $^{23}$Na concentration (0, 20, 40, 60, and 80 mM, respectively) were placed on the participants' limb within the field of view and were used for quantification. All participants were scanned on a 3 T mMR Biograph with a transmit/receive birdcage $^{23}$Na coil (Stark Contrast MRI Coils Research). Three sets of $^{23}$Na weighted MR images were acquired with a balance steady-state free precision (BSSFP) sequence (Bandwidth = 289 Hz/pixel, Echo Time = 3.33 ms, Repetition Time = 21 ms, Signal Averages = 100, resolution = $3.1 \times 3.1 \times 20$mm$^3$, acquisition time = 2:17). Quantified $^{23}$Na maps were calculated with an in-house script (Matlab). Regions of Interests were drawn per acquisition per participant on each sodium phantom and the averaged signal per phantom was used to calculate a calibration curve as: [Known $^{23}$Na Concentration] = p1*[Averaged signal]+p2, where p1 and p2 are the slope and intercept, respectively. The values of p1 and p2 were then used to calculate the $^{23}$Na concentration per pixel using: [$^{23}$Na Concentration in mM] = ([Signal] – p2)/p1.

Mean sodium concentrations in the skin and in 4 muscle groups were determined for each patient. Image analysis was performed using ITK-Snap v3.0 software (http://www.itksnap.org). Mean sodium concentrations in the skin and muscles were compared between SLT patients and controls.

**Urine culture experiments.** To characterise the urinary microbial community, urine specimens were collected from SLT patients ($n = 14$) and HCs ($n = 12$), and sediment cultures undertaken. A 20 ml volume urine specimen was collected and cells pelleted by centrifugation. Cell pellets were resuspended in 400 μl of phosphate buffered saline (PBS) (Life Technologies, UK) and ten-fold serial dilutions were performed in PBS to facilitate the quantification of microbial colonies. Cultures were kept in an ordinary incubator at 37 °C for 18–24 h. Following incubation, the resulting microbial growth was quantified in colony-forming units per millilitre (cfu/ml).

Cultured organisms were grown separately on Columbia blood agar (Oxoid, UK) for species-level identification using matrix-assisted laser desorption ionisation time-of-flight mass spectrometry (MALDI-TOF MS). The direct colony protocol was adopted on the MicroFlex LT mass spectrometer (Bruker Daltonics, USA). Briefly, each cultured isolate was spotted on a 96-well target plate and overlaid with 1 μl matrix solution consisting of alpha-cyano-4-hydroxycinnamic acid dissolved in 50% acetonitrile and 2.5% trifluoroacetic acid. Once air-dried for 5 min, the target plate was loaded and analysed using the MALDI Biotyper 3.0 software programme containing the Bruker Taxonomy 16S reference library. The number of different isolates identified, the microbial burden of each isolate,

**Table 2 Whole blood cytokine stimulation assays to assess innate cytokine responses in salt-losing tubulopathy patients.**

| Stimulation | Salt-losing tubulopathy | Healthy control | P-value |
|---|---|---|---|
| TNF (pg/ml) | | | |
| IFNg | 2.4 (2.4–2.4) | 2.4 (2.4–2.4) | 0.09 |
| LPS | 3166 (1669–5083) | 1420 (928–2654) | **0.0091** |
| LPS/IFNg | 8584 (6225–12,665) | 4945 (2937–6201) | **0.0006** |
| ZYM | 2588 (2051–4714) | 1740 (1352–2722) | **0.0079** |
| ZYM/IFNg | 3328 (2432–6574) | 2137 (1496–2696) | **0.0006** |
| B-GLUC | 1482 (951–2532) | 1107 (500–1630) | **0.029** |
| B-GLUC/IFNg | 2408 (900–4286) | 1539 (940–2223) | 0.09 |
| PAM2 | 229 (137–827) | 73 (34–166) | **0.0007** |
| MDP | 58.8 (21–163) | 13 (7.2–24.9) | **0.0023** |
| IL-6 (pg/ml) | | | |
| IFNg | 1.4 (1.4–16.9) | 1.4 (1.4–1.4) | 0.06 |
| LPS | 20,733 (16,734–28,187) | 14,128 (10,436–18,901) | **0.0025** |
| LPS/IFNg | 26,489 (18,107–32,192) | 15,717 (11,840–23,041) | **0.0038** |
| ZYM | 8370 (6529–12,742) | 5033 (3406–6608) | **0.0057** |
| ZYM/IFNg | 5636 (3113–6686) | 2358 (1424–3117) | **0.0049** |
| B-GLUC | 12,239 (10,664–20,716) | 11,562 (6427–14,381) | 0.17 |
| B-GLUC/IFNg | 7368 (1011–12,052) | 3456 (1982–5628) | 0.16 |
| PAM2 | 5771 (2518–9729) | 1035 (608–3297) | **0.0004** |
| MDP | 236 (50–1977) | 16.1 (7.6–139) | **0.0027** |
| IL-1β (pg/ml) | | | |
| IFNg | 7.1 (1.5–33.2) | 1.08 (1.08–2.92) | **0.0007** |
| LPS | 7047 (3361–9777) | 3344 (2141–4708) | **0.0017** |
| LPS/IFNg | 11.197 (6561–16,324) | 8832 (5805–11,057) | 0.06 |
| ZYM | 18,009 (9639–21,308) | 8880 (7192–11,562) | **0.0061** |
| ZYM/IFNg | 20,704 (7319–26,839) | 8741 (6584–12,962) | **0.004** |
| B-GLUC | 4026 (2126–6700) | 2675 (1570–3984) | 0.051 |
| B-GLUC/IFNg | 8458 (2217–11,370) | 3630 (2229–5578) | **0.032** |
| PAM2 | 581 (177–1367) | 151 (56–325) | **0.0023** |
| MDP | 65 (22–298) | 14 (6.6–50) | **0.0089** |
| IL-10 (pg/ml) | | | |
| IFNg | 0.52 (0.52–0.52) | 0.52 (0.52–0.52) | 0.53 |
| LPS | 771 (521–997) | 882 (589–1207) | 0.42 |
| LPS/IFNg | 265 (129–335) | 227 (117–365) | 0.95 |
| ZYM | 140 (76–262) | 162 (72–237) | 0.98 |
| ZYM/IFNg | 31 (8–87) | 37 (9.7–60.7) | 0.95 |
| B-GLUC | 102 (29–149) | 51 (20–79) | **0.04** |
| B-GLUC/IFNg | 10 (0.8–34) | 3.8 (0.8–7.6) | 0.18 |
| PAM2 | 42 (28–100) | 16.2 (7.2–51.2) | **0.02** |
| IL-12 (pg/ml) | | | |
| IFNg | 5.1 (5.1–5.1) | 5.1 (5.1–5.1) | 0.24 |
| LPS | 33.3 (7.3–40.5) | 7.9 (5.1–25.9) | **0.02** |
| LPS/IFNg | 322 (198–428) | 139 (61–218) | **0.0011** |
| ZYM | 10.9 (6.4–44.2) | 5.1 (5.1–18.2) | 0.07 |
| ZYM/IFNg | 80 (46–113) | 49 (15–113) | 0.1 |
| B-GLUC | 5.1 (5.1–15.8) | 5.1 (5.1–5.1) | 0.22 |
| B-GLUC/IFNg | 18.2 (5.1–54) | 10.6 (5.1–24.9) | 0.27 |
| IFN-γ (pg/ml) | | | |
| IL12 | 5.4 (5.4–5.4) | 5.4 (5.4–5.4) | 0.99 |
| LPS | 5.4 (5.4–5.4) | 5.4 (5.4–5.4) | 0.18 |
| LPS/IL12 | 420.7 (155.8–1161) | 277.7 (138.3–572.8) | 0.35 |
| IL18 | 5.4 (5.4–5.4) | 5.4 (5.4–5.4) | 0.89 |
| IL18/IL12 | 4587 (2792–10,970) | 391.1 (216–730) | 0.28 |
| ZYM | 5.4 (5.4–23.0) | 5.4 (5.4–17.2) | 0.69 |
| ZYM/IL12 | 477.4 (107–866.9) | 391.1 (216–730) | 0.88 |
| B-GLUC | 5.4 (5.4–9.25) | 5.4 (5.4–5.4) | 0.27 |
| B-GLUC/IL12 | 370 (50.25–1060) | 269.6 (94.2–557.1) | 0.93 |
| IL15 | 5.4 (5.4–5.4) | 5.4 (5.4–5.4) | 0.25 |
| IL15/IL12 | 43.1 (9.6–173.2) | 42.55 (19.28–148.1) | 0.98 |

Cytokine concentrations (TNF, IL-1β, IL-6, IL-10, IL-12, and IFN-γ) are reported as median and interquartile range and compared between SLN patients ($n = 13$) and healthy controls ($n = 42$) using a two-sided Mann–Whitney test.
Source data are provided as a Source data file.
*LPS* lipopolysaccharide, *ZYM* zymosan, *B-GLUC* beta-glucan, *MDP* muramyldipeptide.
Significant P values ($P \leq 0.05$) are highlighted in bold.

and the microbial identity of the isolates was recorded for each patient and summarised and compared between cohorts.

To determine if increasing extracellular sodium directly affected microbial growth, strains of *Escherichia coli, Staphylococcus aureus, Candida albicans*, and *Corynebacterium amycolatum* were grown on chromogenic agar (Biomerieux, France) in an ordinary incubator at 37 °C. A bacterial suspension in sterile distilled water with turbidity equivalent to a 0.5 McFarland standard was made for each strain. Seven ten-fold serial dilutions were made for each strain and 25 µl of each dilution, along with the undiluted bacterial suspension, were plated onto one quadrant of four tryptone soya agar plates, which had been prepared with and without additional sodium chloride (unadjusted plate, +40 mM, +80 mM, +160 mM). Plates were incubated at 37 °C and microbial growth was examined by visual inspection after 24 h.

**Initial immunological analysis**. Initial immunological analysis of SLT patients was undertaken in the National Health Service clinical laboratories of the Royal Free Hospital, London. This included full blood count, erythrocyte sedimentation rate (ESR), C-reactive protein (CRP), immunoglobulin A, G, M, and E concentrations, complement protein C3 and C4 concentrations, lymphocyte counts and subsets (CD3$^+$, CD4$^+$, CD8$^+$, CD19$^+$, CD16/56$^+$).

To assess T-cell proliferation and activation, peripheral blood mononuclear cells (PBMCs) were isolated from SLT patients ($n = 6$) and HCs ($n = 7$) by density centrifugation (Lymphoprep; Stemcell, cat#07861), and cells were stimulated for 72 h with anti-human CD3 (10 µg/ml plate bound; BD Pharmigen, cat# 555336) and anti-human CD28 (1 µg/ml; Invitrogen, cat# 16-0289-85) in XVIVO15 media (Lonza, cat# BE02-060F). Cells were stained for viability (Fixable viability dye—efluor450; Invitrogen, cat# 65-0863-14), CD4 (CD4-FITC; BioLegend, cat# 300538), and CD25 (CD25-PE/Cy7; BioLegend, cat# 302612), prior to fixation and permeabilisation (Invitrogen, Cat# 00-5523-00) and intracellular staining for Ki67 (Ki67-APC; BioLegend, cat#350514). The proportion of viable CD4$^+$ cells expressing Ki67 and CD25, analysed by fluorescent activated cell sorting (FACS), are reported. Gating for this and all subsequent FACS experiments was via fluorescence minus one (FMO). Varicella zoster and pneumococcal-specific immunoglobulin G levels in serum were measured via enzyme-linked immunosorbent assay (ELISA) (VaccZyme VZV and PCP kits; Binding site, Cat#MK012 and MK092).

**T-cell subset analysis**. PBMCs were isolated from SLT patients ($n = 20$), HCs ($n = 19$), and disease controls ($n = 18$) and fresh cells were stimulated for 4 h with phorbol myristate acetate (PMA, 50 ng/ml; Sigma, cat#P8139) and ionomycin (1 µg/ml; Sigma, cat#I9657) in the presence of brefeldin A (5 µg/ml). Cells were subsequently stained for CD4 (CD4-FITC), IL-17 (IL-17-PE; BioLegend, cat#512306), IFNγ (IFNγ-APC; BioLegend, cat#502512) and IL-4 (IL-4-PE/Cy7; BioLegend; cat#500824), and analysed by FACS. The proportion of CD4$^+$ cells expressing each of these cytokines was determined and compared between groups. Unstimulated cells were also stained for CD3 (CD3-PE/Cy5; BioLegend, cat#300410), CD4 (CD4-FITC), CD8 (CD8-BV785; BioLegend, cat#301046), CD45RA (CD45RA-AF700; BioLegend, cat#304120), and CD45RO (CD45RO-PE; BioLegend, cat#304206), and T-cell subsets were compared between SLT patients and controls.

To investigate the effect of extracellular ions on CD4$^+$ subset balance, healthy control PBMCs were isolated and stimulated for 72 h with anti-human CD3 (10 µg/ml plate bound) and anti-human CD28 (1 µg/ml) in XVIVO15 media with additional 0–40 mM NaCl, 0–2 mM KCl, and 0–1 mM MgCl$_2$. On day 3, cells were restimulated for 4 h with PMA (50 ng/ml) and ionomycin (1 µg/ml; Sigma, cat#I9657) in the presence of brefeldin A (5 µg/ml). Cells were then stained for viability (eFluor450), CD4 (CD4-FITC), IFNγ (IFNγ-APC), IL-4 (IL-4-PE/Cy7) and IL-17 (IL-17-PE), and analysed by FACS. The ratio of cytokine expression in viable CD4$^+$ cells was determined.

Electrolyte concentrations in unadjusted XVIVO15 media were measured by automated analyser (Cobas 8000 modular analyser series, Roche).

**Th17 and Tc17 polarisation analysis**. PBMCs were isolated from SLT patients ($n = 27$), HCs ($n = 21$) and disease controls ($n = 19$) and fresh cells stimulated with anti-human CD3 (10 µg/ml plate bound), anti-human CD28 (1 µg/ml), IL-1β (12.5 ng/ml; Peprotech, cat#200-01B), IL-6 (25 ng/ml; Peprotech, cat#200-06), IL-21 (25 ng/ml; Peprotech, cat#200-21), IL-23 (25 ng/ml; Peprotech, cat#200-23), and TGFβ (5 ng/ml; Peprotech, cat#100-36E) in XVIVO15 media for 7 days. On day 7, cells were restimulated for 4 h with PMA (50 ng/ml) and ionomycin (1 µg/ml; Sigma, cat#I9657) in the presence of brefeldin A (5 µg/ml). Cells were then stained for viability (eFluor450), CD4 (CD4-FITC), CD8 (CD8-PE/Cy7, BioLegend, cat#300914), and IL-17 (IL-17-PE), and analysed by FACS. Supernatant IL-17 concentrations were measured by ELISA (R&D systems, cat#DY317). The proportion of viable CD4$^+$ and CD8$^+$ cells expressing IL-17, representing Th17 and Tc17 cells, respectively, were determined and compared between groups.

These experiments were also undertaken with the addition of osmoles to culture conditions (NaCl 0–80 mM, KCl 0–2 mM, MgCl$_2$ 0–1 mM, mannitol 0–160 mM, and sodium gluconate 0–80 mM) in addition to experiments with aldosterone (100 nM; Cambridge biosciences, cat#B2153), angiotensin II (0.1–1 µM; Sigma,

cat#A9525), and the sodium chloride cotransporter (NCC) inhibitor hydrochlorothiazide (20 µM; Sigma, cat#H4759). IL-17 responses in supplemented media were compared to standard culture conditions. In experiments undertaken with NaCl, sodium gluconate, and mannitol, supernatant osmolality at the end of the 7-day culture was measured (Osmomat 030, Gonotec, Berlin), and osmolality created by each osmole was correlated to Th17 and Tc17 polarisation. Experiments were also undertaken to assess the effect of NaCl specifically on isolated naive CD4$^+$CD45RA$^+$ cells (EasySep Human Naïve CD4$^+$ T cell Isolation Kit; Stemcell, cat#17515), naive CD8$^+$CD45RA$^+$ cells (EasySep Human Naïve CD8$^+$ T cell Isolation Kit; Stemcell, cat#19258), and memory CD4$^+$CD45RO$^+$ cells (EasySep Human Memory CD4$^+$ T cell Isolation Kit; Stemcell, cat#19117). Cell purity (>90%) was confirmed with FACS analysis.

**STAT1 and STAT3 phosphorylation assays**. To investigate for phosphorylation defects in STAT1 and STAT3, PBMCs from SLT patients ($n = 13$) were isolated and cells were stimulated for 15 min with IL-6 (100 ng/ml), IL-21 (100 ng/ml), and IFNα (11,500 U/ml; R&D Systems, cat#11101-1) according to previously described diagnostic protocols[71,72]. Cells were fixed immediately (Cytofix Fixation Buffer; BD Biosciences, cat#554655), permeabilised (Phosflow Perm Buffer III; BD Biosciences, cat#558050), and stained for CD4 (CD4-FITC), pSTAT1 (STAT1 pY701—Alexa647; BD Biosciences, cat#612597) and pSTAT3 (STAT3 pY705 – PE; BD Biosciences, cat#612569). For each subject, the increase in phosphorylation of STAT1 and STAT3 in stimulated compared to unstimulated CD4$^+$ and CD4$^-$ cells were determined, and this increase was expressed as a ratio with the increase seen in a healthy control subject in each experiment.

**Sodium chloride cotransporter (NCC) expression**. Healthy control PBMCs were isolated and stained for NCC (rabbit anti-SLC12A3 polyclonal primary antibody; Life technologies, cat# PA5-80004; goat anti-rabbit APC conjugated secondary antibody; Invitrogen, cat#A-10931), CD4 (CD4-FITC), CD8 (PE-Cy7), and CD45RA (CD45RA-AF700), and analysed by FACS. Separate staining was performed for NCC (APC), CD45 (CD45-FITC; BioLegend, cat#304006), CD3 (CD3-BV711), and CD56 (CD56–BV510; BioLegend, cat#318340), in addition to NCC (APC) with CD19 (CD19-PE/Cy7; Invitrogen, cat#25-0199042). The percentage of each cell type expressing NCC was determined.

**Calcium flux experiments**. PBMCs were isolated from HCs ($n = 13$) and SLT patients ($n = 5$), and fresh cells loaded with the ratiometric calcium indicator Indo-1 (4 µM; Indo-1 AM cell permeant; ThermoFisher, cat#I1223). Cells were washed and stained for CD4 (CD4-PerCP/Cy5.5; BioLegend, cat#300530) and then incubated with biotinylated anti-CD3 (BioLegend; cat#317320) in XVIVO15 media for 15–30 min. The ratio of Indo-1 fluorescence in CD4$^+$ cells at 400 nM compared to 475 nm (representing free intracellular calcium) was then analysed by FACS. Fluorescence data were collected for 1 min to determine baseline calcium concentration, prior to T-cell activation with the addition of streptavidin (40 µg/ml; Avivasysbio, cat#OPPA01200), and fluorescence recorded for a further 5 min. A calcium flux curve was created and the area under this curve compared between SLT patients and HCs.

**IL-23 receptor expression**. PBMCs were isolated from HCs ($n = 8$) and SLT patients ($n = 7$), and stained for CD4 (CD4-FITC) and the IL-23 receptor (IL23R-PE; R&D Systems, cat#FAB14001P) and analysed with FACS. The percentage of CD4$^+$ and CD4$^-$ cells expressing the IL23-R were determined and compared between groups.

**SGK1 expression**. PBMCs were isolated from HCs ($n = 7$) and SLT patients ($n = 13$) and protein lysates prepared by digestion in RIPA buffer (Sigma, cat#R0278) containing protease inhibitors (Halt protease inhibitor cocktail; ThermoScientific, cat#1862209). Protein was quantified (Pierce BCA assay; ThermoScientific, cat#23227) and 35 µg samples loaded onto a 10% SDS-PAGE electrophoresis gel run at 120 V for 60–90 min (BioRad PowerPac Basic). The gel was transferred (70 V for 90 min) to a nitrocellulose membrane (GE Healthcare, cat#106000006) and blocked for 1 h at room temperature in 5% milk. The membrane was incubated with primary antibody at 4 °C overnight and secondary antibody for 1 h at room temperature. Rabbit anti-SGK1 primary antibody (Abcam, cat#ab43606) was used at 1:900 dilution and mouse anti-GAPDH primary antibody (Abcam, cat#ab8245) at 1:10,000 dilution. Anti-rabbit HRP (Abcam, cat#ab205718) and anti-mouse HRP (Abcam, cat#ab205719) secondary antibodies were used at 1:3000 and 1:5000 dilutions, respectively. Blots were developed with the BioSpectrum 810 Imaging System (UVP) and images quantified using ImageJ software (https://imagej.net). The expression of SGK1 relative to that of GAPDH was determined and compared between SLT patients and HCs.

**NFAT5 expression**. PBMCs were isolated from HCs ($n = 13$) and SLT patients ($n = 12$) and RNA extracted using an RNA purification kit (Direct-zol RNA MicroPrep; Zymo Research, cat#R2061). Yields were analysed using nanodrop spectrophotometry and complementary DNA synthesis undertaken using a high capacity cDNA reverse transcription kit (Applied Biosystems,

cat#4374966) using a G-Storm GS1 thermal cycler. Reverse Transcriptase Polymerase Change Reaction (RT-PCR) was then performed using Taqman reagents (Taqman Fast Advanced Master mix; Applied Biosystems, cat#4444556) in the LightCycler96 (Roche) thermal cycler. Primers were used to detect NFAT5 (Taqman gene expression assay ID Hs00232437_m1; Applied Biosystems, cat#4331182) and 18S (Taqman gene expression assay ID Hs99999901_s1; Applied Biosystems, cat#4331182). The expression of NFAT5 relative to 18S in each subject was determined using $2^{-\Delta CT}$ and then compared to one healthy control subject using $2^{-\Delta\Delta CT}$. Relative expression was compared between SLT patients and HCs.

**Innate cytokine stimulation assays**. Whole blood of SLT patients ($n = 13$) and HCs ($n = 42$) was diluted 1:5 in RPMI medium and activated in 96-well plates with the following stimuli used alone or in combination: IFNγ ($2 \times 10^4$ IU/ml, Immuno Tools); IL-12 (5 µg/ml; Immuno Tools); Lipopolysaccharide (LPS, 1 µg/ml; List Biochemicals); IL-18 (5 µg/ml; R&D Systems); Zymosan (ZYM, 10 mg/ml; InvivoGen); Beta-glucan (B-GLUC, 5 mg/ml, InvivoGen); IL-15 (5 µg/ml; Immuno Tools); Pam2 (1 mg/ml; EMC Microcollections); Muramyldipeptide (MDP, 4 mg/ml; InvivoGen).

Cells were incubated for 24 h and supernatant cytokines then measured by multiplex bead array (TNF, IL-1β, IL-6, IL-10, IL-12, and IFN-γ; R + D Systems Fluorokinemap) on a Luminex analyzer (Bio-Plex, BioRad, UK). Cytokine concentrations were compared for each stimulating condition between SLT patients and HCs.

**Monocyte analysis**. PBMCs were isolated from SLT patients ($n = 14$) and HCs ($n = 12$) and stained for CD14 (CD14-APC/Cy7; BioLegend, cat#325619) and CD16 (CD16-FITC; BioLegend, cat#302006) and analysed with FACS. Classical (CD14$^+$CD16$^-$), intermediate (CD14$^+$CD16$^+$) and non-classical (CD14$^-$CD16$^+$) monocyte populations were determined and compared between groups. Cells were also stimulated for 4 h with LPS (100 ng/ml; LPS from *E. coli* Serotype R515; Hycult Biotech, cat#HC4048) in the presence of Brefeldin A (5 µg/ml). Cells were stained for viability (alexafluor450) and CD14 (APC/Cy7), prior to being fixed and permeabilised, stained for TNF (TNF-Alexafluor488; BD Biosciences, cat#557722) and analysed with FACS. This was done in standard media and, in HCs, in media supplemented with 40 mM NaCl, 2 mM KCl, 1 mM MgCl$_2$, and aldosterone (10–100 nM). The proportion of viable CD14$^+$ cells expressing TNF was compared between SLT and controls, and between supplemented media and standard conditions.

**NK cell analysis**. PBMCs were isolated from SLT patients ($n = 16$) and HCs ($n = 17$), and fresh cells were stained for CD45 (CD45-FITC), CD3 (CD3-BV711), CD56 (CD56–BV510), and CD16 (CD16-PE/Cy7; BioLegend, cat#302016), and analysed via FACS. CD45$^+$CD3$^-$CD56$^+$CD16$^+$ (NK) cells as a proportion of CD45$^+$ cells were calculated, as were the ratio of CD56$^+$CD16$^+$ to CD56$^+$CD16$^-$ cells. PBMCs were also stimulated with IL-12 (50 ng/ml; Peprotech, cat#200-12) and IL-18 (50 ng/ml; Cambridge bioscience, cat#230-00229-10) in the presence of brefeldin A (5 µg/ml; Sigma, cat#B7651) for 4 h and stained for viability, CD45, CD3, CD56, and IFNγ. The proportion of viable CD45$^+$CD3$^-$CD56$^+$ cells expressing IFNγ was determined. Stimulation was also performed in HCs ($n = 8$) in media supplemented with 40 mM NaCl. The proportion of NK cells (CD45$^+$CD3$^-$CD56$^+$) was determined as was the expression of IFNγ by CD56$^+$ cells, and compared between standard and high salt conditions.

**Antibodies**. Antibodies used are outlined in Supplementary Table 10.

**FACS analysis**. FACS was undertaken on a LSR Fortessa flow cytometer (BD Biosciences). The analysis was undertaken using FlowJo v10 software (https://www.flowjo.com). Gating strategies are outlined in Supplemental Fig. 11.

**Statistics**. Data are presented as number and percentages for categorical variables, and mean and standard deviation or median and interquartile range for numerical variables depending on the distribution. Categorical variables were compared using the Fisher's exact or Chi-squared test. Non-parametrically distributed numerical variables were compared between two groups using the Mann–Whitney test and the Wilcoxon test for unpaired and paired data, respectively. Parametrically distributed numerical variables were compared with an unpaired or paired *t*-test. Variables are compared across greater than two groups with a one-way analysis of variance with multiple comparison testing; specifically, for non-parametric data, we used the Kruskal–Wallis test with Dunn's test for multiple comparisons. Numerical variables were correlated with a Pearson correlation. Unless otherwise specified, graphs plot individual points alongside median and interquartile range.

Analysis was performed using Graphpad Prism version 7 (www.graphpad.com). A *p*-value of ≤0.05 was considered statistically significant.

**Study approval**. The Royal Free Hospital Research and Development committee approved the study (identification number 7727) and all participants provided written informed consent prior to enrolment. Consent was also obtained to publish patient identifiable images where appropriate.

**Reporting summary**. Further information on research design is available in the Nature Research Reporting Summary linked to this article.

## Data availability
The data that support the findings of this study are outlined in the Data Source File. Additional data including original flow cytometry files are freely available from the corresponding authors upon reasonable request. Source data are provided with this paper.

## Code availability
The code used in the analysis of $^{23}$Na-MRI is freely available from the corresponding authors upon reasonable request.

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

## Acknowledgements

We are truly thankful to the participants of the study. We acknowledge Kidney Research UK (Ref: TF_007_20161125; Evans) and the Medical Research Council (Ref: MR/P001777/1; Antonelou) for their funding and support.

## Author contributions

R.D.R.E., S.B.W., and A.D.S. designed the study; R.D.R.E., M.A., S.S., M.R., S.H., L.C.-G., G.B.-M., C.A.M., and RD conducted the experiments and analysed the data; R.D.R.E., S. B.W., and A.D.S. drafted and revised the manuscript; all authors approved the final version of the manuscript.

## Competing interests

We authors declare no competing interests.
