## [Peer Review File · Nature Communications]

REVIEWER COMMENTS

Reviewer #1 (Autoimmunity and immunophenotyping, Th17) (Remarks to the Author):

The central claim of this innovative paper is that chronic extracellular depletion of sodium, potassium, and magnesium, as seen in patients with inherited salt losing tubulopathies (SLT), impacts T cell polarization and impairs IL-17 responses, leading to immunodeficiency. By characterizing the immune profile of patients with SLT and conducting extensive analyses and in vitro experiments, the authors succinctly illustrate a novel mechanism of IL17 immunodeficiency mediated by an altered extracellular ionic environment.

The present work is convincing because the authors use multiple methodologies to produce robust, consistent findings. First, the paper presents a detailed characterization of the SLT clinical immunophenotype that parallels the pattern of mucosal infections and autoimmune disease seen in patients who have IL-17 deficiencies secondary to STAT1/3 deficiencies. After confirming the presence of an IL-17 deficiency at a cellular level, the authors then present a series of experiments which illustrate that the ionic environment, rather than STAT1/3 phosphorylation, calcium flux, or sodium transporter function, is the cause of this deficiency.

This paper represents the first in-depth characterization of dysregulated immunity in patients with inherited SLT. Highlighting this previously unrecognized component of SLT will impact how these diseases are understood and treated. Na-MRI imaging also shows that SLT patients are missing the sodium stores present in the muscle and skin of healthy people, a finding which may provide context for future translational research on the role of local tissue environments in immune responses. While excess sodium promoting Th17 responses has been well-documented, this paper shows that sodium depletion leads to deficient Th17 responses and IL-17-mediated immunodeficiency. The authors also highlight a potential effect of sodium on Th2 and NK responses, and suggest that other cations, such as potassium and magnesium, are influential in Th17 polarization. These findings will be of interest to basic and translational researchers.

Major comments:

1. More information pertaining to how healthy controls differ from disease controls is needed. Clinical data for these individuals would be beneficial. Are clinical scales and associated syndromes being devised based on medical records? Such clinical data should be succinctly and methodically presented at outset.
2. Line 183. It is unclear the utility of providing data pertaining to urinary microbial differences as this is not incorporated later in the text.
3. Line 201. Title does not take into account discussion about T cell function.
4. Line 209. Ki67 is used as a proxy for proliferation. Authors could consider using anti-CD3 and anti-CD28 stimulation over a 3-5 day period with CFSE and then assessing proliferation.
5. Line 213. Varicella zoster and pneumococcal are used to comment on antigen specific antibody responses and presumed B cell activity, however, this is not commented on later in text and is also not entirely reflective of humoral function. Authors may wish consider memory B cell in vitro culture with CpG, CD40L, IL-21, IL-6 and subsequent immunoglobulin G ELISA analysis.
6. Analysis of Th17/Tc17/Th1/Th2 subsets and follow-up experiments is based on whole PBMC populations or magnetic separation. FACS cell sorting is not pursued to improve isolation of cellular populations ie: CD4+, CD8+, naive CD45RA+, and memory CD45RO+ to better analyze for difference between patients and controls. This would be preferred. (PMID: 23467095)
7. Data is presented as ratios among bulk CD4+ populations raising concern for masking of lack of differences in cytokine production in independent populations as described. In the absence of polarizing (naïve) CD4+CD45RA+ cells to Th1/Th2/Th17 phenotypes, the expression of linked

cytokines does not equate to the distinct populations themselves and renders conclusions incomplete.

8. STAT1 and STAT3 analysis is done in bulk cell populations raising concerns identical to those above.
9. Authors should comment on perceived changes due to magnesium concentration shifts when tables do not suggest differences between patients and controls.
10. Given aforementioned, overwhelmingly, conclusions seem grandiose and without sufficient data to support them.

Minor comments:

11. Figures appear disjointed.
 - a. Figure 1A should be of higher quality.
 - b. P values are inconsistently presented and are not reflected in figures and scattered around legends.
 - c. Scale bars are not consistent throughout and when compared in subfigures should read the same across.
 - d. Cell density isn't presented.
 - e. Are flow cytometry dot plots representative or are they combination of samples? One might assume they are representative dot plots, but they should be explicitly described as such.
 - f. Dot plots could be labelled in a manner that would make it easier for a reader to understand populations/axes.
 - g. Dot plots also appear grainy and intensity isn't visible. It should be presented universally throughout main and supplemental figures.

Reviewer #2 (Hypertension-immune crosstalk) (Remarks to the Author):

The current studies examine changes in T lymphocyte polarization associated with salt-wasting tubulopathies (SLTs). 47 patients with SLTs are examined. These patients have increased prevalence of infection and allergic phenomena, altered urinary organisms, increased circulating Th2/Th17 lymphocyte ratios, reduced Th17 cell polarization with ex vivo restimulation, elevated levels of innate immunity cytokines, and NK cells showing blunted IFN-gamma production. The analysis is comprehensive. The studies do not elucidate new pathways through which extracellular sodium loss limits Th17 responses, but rather provide an important and compelling confirmatory observation in a unique patient population. As the epithelial sodium channel (ENaC) has been shown to impact immune cell activation in the setting of hypertonicity, the authors are asked to examine the impact of amiloride on T cell polarization in their patients. The authors find that kidney function inversely correlates with infection scores. The authors are asked to examine the relationship between blood pressure and infection as it seems plausible that sodium loss would lead both to lower blood pressures and higher rates of infection. Last, as interstitial sodium is thought to impact dendritic cell priming, the authors are asked to provide data if available regarding the capacity of the patients' dendritic cells to activate T lymphocytes.

Reviewer #3 (Salt metabolism) (Remarks to the Author):

General Comments: Evans and colleagues show that immune cells from patients with mutations of electrolyte transporters display reduced Th17 polarization, which can be reversed in vitro by increasing the NaCl concentration in the cell culture medium by 40 mmol/L. Because this immune cell phenotype is associated with a higher incidence of infections, and because some of the patients show lower skin Na content, the authors conclude that chronic (renal) salt wasting associates with immunodeficiency due to an altered extracellular ionic environment that impacts T cell polarization. To my opinion, this is an important clinician-scientist study which capitalizes on a relevant and carefully

characterized cohort of patients with electrolyte transporter mutations. My major criticism is that the patho-physiological connection between renal effects, skin microenvironmental effects, and immune cell function that the authors construct is not (yet) substantiated by the data.

Specific Comments:

1. General Approach – Physiology:

(i) The authors construct the view that chronic salt loading leads to “non-osmotic sodium storage in muscles and the skin, resulting in interstitial sodium concentrations at these sites in excess of those in plasma,” and that this sodium storage will activate immune cells. This construct is not helpful. In their in-vitro experiments, the authors increase the Na concentration in the cell culture medium by 40 mmol/L and thereby induce an osmotic stress reaction with osmotically active sodium. Indeed, many of the described hypertonicity-driven immune cell responses take place in-vivo in the skin at tissue (Na+K) / tissue water ratios of approximately 190 mmol/L, suggesting the presence of a hypertonic environment. Thus, the authors may want to prevent the traditional term “non-osmotic sodium storage”, which derives from an era where investigators believed that the sodium concentrations in blood and in peripheral extracellular fluids are not much different, and that relevant osmotic gradient formation only exists in the kidney. This view is most likely not correct; the skin might alternatively be viewed as an osmotic gradient-generating, kidney-like counter current system (Pflugers Arch. 2015 Mar;467(3):551-8).

(ii) In line with the above comments, it is unclear to me why the authors conclude from the data they present that the reduction in skin Na concentration in patients with mutations of electrolyte transporters are secondary to renal salt wasting: in Supplemental Table 2, the authors report normal serum Na concentrations in their patients, while in Figure 1 they report a reduction in skin Na concentration. My conclusion is that the patient’s kidneys have achieved constancy in the plasma Na microenvironment, while the interstitial Na concentration specifically in the skin (not in muscle) is reduced, suggesting an extrarenal problem (keratinocyte transport?). My understanding after having had a quick look at the tissue-specific expression of reported mutated genes in the patients (Supplemental Table 1) at <http://biogps.org/#goto=welcome> is that at least the described transporter mutations in patients with Gitelman and EAST syndromes are not kidney specific. I conclude that MRI analysis of potential differences in the skin sodium storage phenotype between patients with Bartter syndrome and patients with Gitelman or EAST syndromes might provide with important additional insights that could further substantiate the authors hypotheses on the importance of the ionic microenvironment for immunological host defense.

2. General Approach – Immunology:

Similarly, patients with Bartter, Gitelman, or EAST syndromes may show different levels of immune cell activation, because their immune cells may express different levels of mutated transporters in their cell membranes (<http://biogps.org/#goto=welcome>). This may be relevant for host defense mechanisms, as well as for renal and extrarenal maintenance of local salt and water homeostasis.

3. Data presentation – Fig. 1:

The authors show a convincing reduction in skin Na content in “SLT” patients. However, the grouping of “SLT” patients is not convincing, because it is not clear whether “SLT” refers to patients with Bartter, Gitelman, or EAST syndrome (n=4). I have 3 specific questions:

1. In patients with Bartter syndrome, one would assume a quite kidney-specific effect of the SLC12A1 mutation, and not much effect in keratinocytes (see comment 1). Is skin Na content in patients with Bartter syndrome different from controls, and/or different from patients with EAST or Gitelman syndrome?

2. In patients with Gitelman syndrome, one would assume a less kidney-specific effect, and an additional effect of the SLC12A3 mutation on keratinocyte transport, which might be mediated by

dendritic cell function (see comments 1 and 2). Is skin Na content in patients with Gitelman syndrome different from controls, and/or different from patients with Bartter or EAST syndrome?

2. In patients with EAST syndrome, one would assume that extrarenal effects of the mutated transporter, including keratinocyte transport, are at least as relevant as renal-tubular effects (see comments 1 and 2). Is skin Na content in patients with EAST syndrome different from controls, and/or different from patients with Bartter or Gitelman syndrome?

4. Data presentation – Fig. 2, IL-17-related infection scores.

Again, the data presentation suggesting that the observed changes are secondary to renal tubular transporter problems, may be misleading. Given the fact that KCNJ1 mutations in patients with EAST syndrome are not really kidney-specific, that SLC12A3 mutations in patients with Gitelman syndrome may also be pretty DC-specific, while SLC12A1 mutations in patients with Bartter syndrome may be much more kidney-specific: are there differences in Th17-related infection scores between patients with EAST, Gitelman, or Bartter syndromes? Online Supplemental Figure 6C suggests that patients with EAST and Gitelman syndromes indeed have more infections. Why is this finding buried in the Online Supplement? Could it be that these patients have higher infection scores compared to patients with Bartter Syndrome, because (i) skin Na contents are different? (ii) because KCNJ1 mutations (EAST) or SLC12A3 mutations (Gitelman) in immune cells reduce their ability to ward-off infections, compared to patients with Bartter syndrome, in whom SLC12A1 is not relevant for modulating immune cell function?

5. Data presentation – Fig. 4 and Fig. 8, reduction in Th17 polarization patterns; and recovery by increasing the salt concentration in the cell culture medium.

In line with comment 4, are there differences in Th17-dependent polarization patterns between cells from patients with EAST, Gitelman, or Bartter syndromes? Does NaCl-driven recovery depend on differences in transporter mutation?

6. Discussion: Data interpretation – “role of channel/transporter activity on immune cells an unlikely reason for altered immunity.”

Given the fact that patients with Gitelman syndrome may express large amounts of mutated SLC12A3 specifically in dendritic cells (<http://biogps.org/#goto=genereport&id=6559>), I wonder whether the authors may have overlooked the role of DCs in this context. DC residing in the low-salt interstitium in patients with Gitelman syndrome may reduce their co-stimulatory signals, resulting in a generally reduced IL17 response in T cells.

7. Discussion:

Data interpretation – the idea that these exciting environmental changes in the skin reported here could be tackled by increasing daily salt consumption may be naïve. We meanwhile know that extrarenal, local activation of salt retaining mechanisms are key for the induction of Na storage, and that this Na storage does not readily respond to changes in dietary salt intake. From a balance-point-of-view, this is not very surprising. A human keratinocyte is exposed to an estimated $5 \text{ L/min} \times 7200 \text{ min} \times 9 \text{ g/L} = 64.8 \text{ kg}$ salt per day, which is offered to the cell via the blood circulation. Do the authors really believe that a 0.006 – 0.020 kg/d variation in salt intake will significantly change this local salt exposure?

Sincerely,

Jens Titze

Remarks to author

Reviewer #1 (Autoimmunity and immunophenotyping, Th17) (Remarks to the Author):

The central claim of this innovative paper is that chronic extracellular depletion of sodium, potassium, and magnesium, as seen in patients with inherited salt losing tubulopathies (SLT), impacts T cell polarization and impairs IL-17 responses, leading to immunodeficiency. By characterizing the immune profile of patients with SLT and conducting extensive analyses and in vitro experiments, the authors succinctly illustrate a novel mechanism of IL17 immunodeficiency mediated by an altered extracellular ionic environment.

The present work is convincing because the authors use multiple methodologies to produce robust, consistent findings. First, the paper presents a detailed characterization of the SLT clinical immunophenotype that parallels the pattern of mucosal infections and autoimmune disease seen in patients who have IL-17 deficiencies secondary to STAT1/3 deficiencies. After confirming the presence of an IL-17 deficiency at a cellular level, the authors then present a series of experiments which illustrate that the ionic environment, rather than STAT1/3 phosphorylation, calcium flux, or sodium transporter function, is the cause of this deficiency.

This paper represents the first in-depth characterization of dysregulated immunity in patients with inherited SLT. Highlighting this previously unrecognized component of SLT will impact how these diseases are understood and treated. Na-MRI imaging also shows that SLT patients are missing the sodium stores present in the muscle and skin of healthy people, a finding which may provide context for future translational research on the role of local tissue environments in immune responses. While excess sodium promoting Th17 responses has been well-documented, this paper shows that sodium depletion leads to deficient Th17 responses and IL-17-mediated immunodeficiency. The authors also highlight a potential effect of sodium on Th2 and NK responses, and suggest that other cations, such as potassium and magnesium, are influential in Th17 polarization. These findings will be of interest to basic and translational researchers.

We thank the reviewer for these encouraging comments

Major comments:

1. More information pertaining to how healthy controls differ from disease controls is needed. Clinical data for these individuals would be beneficial. Are clinical scales and associated syndromes being devised based on medical records? Such clinical data should be succinctly and methodically presented at outset.

Thank you for drawing our attention to this. Disease controls were patients attending the tubular disorders clinic with a diagnosis that was not associated with renal salt wasting (i.e. non salt losing tubulopathy). Such patients were recruited from the same clinics as the SLT patients, and they underwent the same structured clinical history and biochemical analysis. Importantly, the disease controls had significant differences in serum biochemical parameters compared to SLT patients (see supplementary table 4).

Healthy controls were age and sex matched volunteers who worked within the hospital and university, and who had no known medical conditions. Healthy controls underwent structured clinical history in the same manner as the SLT patients and disease controls.

The process of recruitment and analysis of the control groups has been clarified in the methods (section “Salt Losing Tubulopathy and control cohorts”):

“SLT patients underwent a structured clinical history focused on clinical features of dysregulated immunity. This was also undertaken in age and sex matched healthy (n=24) and disease (n=22) controls. Disease controls included patients attending the Tubular Disorders Clinics, but who had no evidence of a significant salt wasting phenotype. **These patients were recruited from the same clinics as the SLT patients, and they underwent the same structured clinical history and biochemical analysis.** Conditions included: proximal tubulopathy = 7; monogenic hypertension = 5 (familial hyperkalaemic hypertension = 3; STX16 mutation = 2); distal renal tubular acidosis/nephrocalcinosis = 3; isolated renal magnesium wasting = 1; hyperparathyroidism = 1; tubulointerstitial nephritis = 2; medullary sponge kidney = 1; diabetes insipidus = 1; non renal hypokalaemia = 1 **(Supplementary Table 2).** **Healthy controls were volunteers who worked within the hospital or university and who had no known medical conditions.** The prevalence of each clinical variable was compared between SLT and control groups.”

Within the results (section 1: SLT Cohort) the following has been added:

“We investigated immunity in 47 patients with genotyped SLT and 46 age-matched healthy and disease controls^{29,30}. 23 (49%) patients had BS, 22 (47%) patients had GS, and 2 (4%) patients had EAST syndrome **(Supplemental Table 1).** **Disease controls were patients attending the tubular disorders clinic with a diagnosis that was not associated with renal salt wasting (Supplementary Table 2).**”

Moreover, a supplementary table (**supplementary table 2**) has been added outlining the demographic and clinical data of the disease controls, as below.

Supplemental Table 2: Demographic and clinical data (diagnosis and serum biochemistry) of disease controls

FHH – familial hyperkalaemic hypertension; dRTA – distal renal tubular acidosis; TIN – tubulointerstitial nephritis

Number	Age	Sex	Diagnosis	Na (mmol/l)	K (mmol/l)	Cl (mmol/l)	HCO ₃ (mmol/l)	Creatinine (umol/l)	cCa (mmol/l)	PO ₄ (mmol/l)	Mg (mmol/l)
1	35	M	Proximal tubulopathy (fumarate toxicity)	141	4.4	103	23	79	2.29	1.22	
2	38	F	Proximal tubulopathy (HNF4A mutation)	137	3.9	102	16	211	2.4	1.03	0.93
3	48	M	Proximal tubulopathy (Lowe syndrome)	142	4.4	102	22	162	2.39	0.87	0.98
4	53	M	Proximal tubulopathy (Wilson disease)	141	4.5	103	22	111	2.49	1.13	0.87
5	33	F	Proximal tubulopathy (fumarate toxicity)	143	3.9	105	23	81	2.36	0.77	1
6	46	M	Proximal tubulopathy (Wilson disease)	141	4.5	99	25	152	2.39	1.26	0.91
7	51	M	Proximal tubulopathy (inherited)	141	4	102	25	120	2.51	0.81	0.93
8	34	F	FHH	141	5	102	25	62	2.4	1.12	0.77

9	60	F	FHH	137	6.4	108	19	59	2.36	1.16	0.75
10	6	M	FHH								
11	23	F	dRTA	140	4.2	103	24	52	2.33	0.93	0.87
12	8	M	dRTA								
13	55	F	dRTA	138	4.7	104	21	51	2.23	0.85	0.85
14	20	F	TRPM6 mutation (isolated Mg wasting)	141	4.1	106	22	42	2.36	0.72	0.51
15	53	M	Primary hyperparathyroidism	142	4.2		25	93	2.49	0.92	0.89
16	49	F	STX16 mutation (pseudohypoparathyroidism)	140	4.1	98	25	62	2.43	1.19	0.81
17	19	M	STX16 mutation (pseudohypoparathyroidism)	140	4.7	104	23	78	2.24	1.66	0.89
18	59	F	TIN	138	4.3	100	28	118	2.38	1.08	0.87
19	51	F	TIN	138	4.2	103	26	145	2.42	1.1	0.76
20	75	M	Medullary sponge kidney	144	4.2	103	25	94	2.39	1.11	0.79
21	15	M	Nephrogenic diabetes insipidus								
22	31	F	Hypokalaemia (resolved) – extra renal loss	142	4.4	101	27	76	2.45	1.16	0.85

2. Line 183. It is unclear the utility of providing data pertaining to urinary microbial differences as this is not incorporated later in the text.

Thank you for asking us to clarify the reasoning behind this analysis.

We initially highlighted that SLT patients had increased mucosal infections (including urinary tract infections) compared to controls based on patient reported medical history, but wanted to support this with objective evidence of an altered mucosal microbiome. Hence, we were interested to understand if there were differences in the microbial community in SLT patients at one of the common sites of infection (genitourinary tract) and determine what these differences were. We demonstrated that there was indeed a difference in the urinary microbial community (see supplementary table 7) with an increase in *Corynebacterium sp.* in SLT patients compared to controls. We went on to determine whether this was due to a direct effect of an altered extracellular ionic environment on the microbe itself. Altering extracellular Na⁺ had no effect on *Corynebacterium sp.* or other bacterial growth. These data support a primary defect in immunity as opposed to a direct anti-microbial effect underlying the altered microbiome and increased infection rate in SLT patients.

The results (Section 3. SLT patients have an altered urinary microbial community) have been altered as below to clarify the reasons for this analysis and the implications of the findings:

“Given the increased prevalence of SLT mucosal microbial infections, we were interested to understand if there were differences in the microbial community in SLT patients at one of the common sites of infection. Hence, we performed urine sediment cultures, which demonstrated there were altered bacterial constituents with an increase in the presence of *Corynebacterium sp.* in SLT patients compared to controls (**Supplemental tables 7 and 8**). To determine whether alterations in extracellular sodium concentration directly affect microbial growth, strains of *C. amycolatum* as well as other commonly encountered microbes (*E. coli*, *S. aureus*, and *C. albicans*) were cultured in media supplemented with NaCl. NaCl had no effect on the growth of the bacteria tested, whereas there was a dose-dependent reduction in the size of *Candida* colonies with additional NaCl (**Supplemental figure 2**). A direct effect of altered extracellular sodium may therefore contribute to mucosal fungal infection; however, the lack of an effect of an altered extracellular environment on bacterial growth supports a primary defect in immunity underlying the altered microbiome and increased urinary infection rate observed in SLT.

3. Line 201. Title does not take into account discussion about T cell function.

Thank you for this suggestion. The Title has been updated:

“Initial immunological analysis of SLT patients suggests a quantitative defect in NK cells, but no defect in T cell activation or proliferation.”

4. Line 209. Ki67 is used as a proxy for proliferation. Authors could consider using anti-CD3 and anti-CD28 stimulation over a 3-5 day period with CFSE and then assessing proliferation.

We thank the reviewer for proposing the use of CFSE to determine proliferation. We stimulated PBMCs with anti-CD3 and anti-CD28 for 72 hours as the reviewer suggests and assessed proliferation by staining for Ki67. Lastovicka et al, Human Immunology, 2016 (<https://www.sciencedirect.com/science/article/pii/S0198885916304360>) have previously demonstrated good correlation between Ki67 and CFSE when assessing proliferation (correlation coefficient = 0.77) and hence we don't feel CFSE assessment is additive. Moreover, as highlighted in this paper, CFSE is known to exert a concentration dependent negative influence on lymphocyte viability and hence Ki67 was used. With this method, we found no differences in T cell proliferation between SLT patients and controls.

5. Line 213. Varicella zoster and pneumococcal are used to comment on antigen specific antibody responses and presumed B cell activity, however, this is not commented on later in text and is also not entirely reflective of humoral function. Authors may wish consider memory B cell in vitro culture with CpG, CD40L, IL-21, IL-6 and subsequent immunoglobulin G ELISA analysis.

We thank the reviewer for raising the issue of defective B cell function, which may underlie immunodeficiency. As outlined in Oliveira et al, J Allergy Clin Immunol, 2010, (<https://www.ncbi.nlm.nih.gov/pmc/articles/PMC3412511/>) and according to published clinical guidelines (<https://primaryimmune.org/wp-content/uploads/2015/03/2015-Diagnostic-and-Clinical-Care-Guidelines-for-PI.pdf>), the initial screening tests used in routine clinical practice to exclude defects in B cell function include an assessment of total immunoglobulin A, G, M, and E, and antigen specific responses to prior infections/vaccines. This screening approach was therefore undertaken in the SLT cohort, which showed no evidence of impaired total or antigen specific antibody production. We have since undertaken an assessment of B cell subsets in one of the SLT patients, as outlined in the table below. There was no defect in the ability to switch from naïve to memory B cells. We agree the experiments suggested by the reviewer would provide a more in depth assessment of B cell function but given that screening tests for B cell function were normal, this more detailed assessment of B cells was not undertaken.

B cell subset	Reference range	SLT patient
---------------	-----------------	-------------

CD19%	4.9-18.4	3.9
CD27-IGD+ (NAÏVE) % OF CD19	42.6-82.3	33.96
CD27+IGD+ (IGM MEMORY) % OF CD19	7.4-32.5	14.62
CD27+IGD- (SWITCHED MEMORY) % OF CD19	6.5 - 29.1	40.57
CD27+IGD- (SWITCHED MEMORY) % OF PBL		1.58
CD21-CD38- % OF CD19	0.9-7.6	25
TRANSITIONAL B CELLS	0.6-3.4	2.36
PLASMA BLASTS	0.4-3.6	7.55

6. Analysis of Th17/Tc17/Th1/Th2 subsets and follow-up experiments is based on whole PBMC populations or magnetic separation. FACS cell sorting is not pursued to improve isolation of cellular populations ie: CD4+, CD8+, naive CD45RA+, and memory CD45RO+ to better analyze for difference between patients and controls. This would be preferred. (PMID: 23467095)

and

7. Data is presented as ratios among bulk CD4+ populations raising concern for masking of lack of differences in cytokine production in independent populations as described. In the absence of polarizing (naïve) CD4+CD45RA+ cells to Th1/Th2/Th17 phenotypes, the expression of linked cytokines does not equate to the distinct populations themselves and renders conclusions incomplete.

We thank the reviewer for raising the issue of differences in T cell populations, and the effect of the extracellular ionic environment on each of these. As outlined in our paper, we propose immunity is altered in SLT patients as a result of their unique extracellular environment and its impact on IL-17 responses in multiple T cell subtypes. Initial work investigating the effect of sodium on Th17 cell polarisation used FACS isolated naïve and memory CD4+ cells as the reviewer refers to (Kleinewietfeld et al, 2013). In these studies, the investigators report that sodium predominantly affected naïve as opposed memory cell populations. Hence alterations in the proportion of naïve and memory cell subsets in SLT patient might explain their altered IL-17 responsiveness. As well as measuring lymphocyte subsets in SLT patients and comparing these to a laboratory reference range (**supplemental table 9**), we have now performed a further analysis of T cell subsets (CD4+, CD8, CD45RA, and CD45RO as the reviewer suggests) in SLT patients and compared cell proportions to age matched healthy controls. There were no differences in the relative frequency of naïve and memory cell populations in SLT patients and controls and hence this does not underlie their altered IL-17 responsiveness.

The following has been added to the results (Section 4):

“...similar to the healthy adult population (**Figure 2F**). Moreover, CD4, CD8, CD45RA, and CD45RO subsets were not different in SLT patients compare to controls (**Supplemental figure 3**).”

A supplemental figure (supplemental figure 3) has been added, as outlined below, detailing these results.

Supplemental Figure 3: T cell subsets in SLT patients and healthy controls

- A. **CD3+CD4+ cells** (% of CD3+ cells): HC 62.2% (52.5-70.4), SLT 71.2% (57.8-73.9), p=0.23. Compared with Mann-Whitney test.
- B. **CD3+CD8+ cells** (% of CD3+ cells): HC 33.1% (25.5-42.4), SLT 26.0% (21.8-35.4), p=0.15. Compared with Mann-Whitney test.
- C. **CD4:CD8 ratio**: HC 1.9 (1.2-2.8), SLT 2.7 (1.6-3.4), p=0.19. Compared with Mann-Whitney test
- D. **CD3+CD45RA+ cells** (% of CD3+ cells): HC 41.0% (29.7-47.8), SLT 32.9% (26.2-58.4), p=0.90. Compared with Mann-Whitney test.
- E. **CD3+CD45RO+ cells** (% of CD3+ cells): HC 43.5% (32.5-56.6), SLT 48.9% (28.1-60.0), p=0.72. Compared with Mann-Whitney test.
- F. **CD3+CD4+CD45RA+ cells** (% of CD3+CD4+ cells): HC 31.3% (17.5-40.2), SLT 28.0% (18.4-51.8), p=0.99. Compared with Mann-Whitney test.
- G. **CD3+CD4+CD45RO+ cells** (% of CD3+CD4+ cells): HC 48.9% (44.1-68.9), SLT 49.4% (30.6-64.6), p=0.69. Compared with Mann-Whitney test.
- H. **CD3+CD8+CD45RA+ cells** (% of CD3+CD8+ cells): HC 66.5% (48.5-77.8), SLT 53.0% (38.9-74.9), p=0.38. Compared with Mann-Whitney test.
- I. **CD3+CD8+CD45RO+ cells** (% of CD3+CD8+ cells): HC 27.9% (18.3-44.9), SLT 36.8% (24.1-55.7), p=0.40. Compared with Mann-Whitney test.

ns – not significant (p>0.05), *p≤0.05, **p≤0.01, ***p≤0.001, ****p≤0.0001
SLT – salt losing tubulopathy; HC – healthy control

A.

B.

C.

D.

E.

F.

G.

The methods have been updated accordingly (Section: T cell subset analysis):

“...The proportion of CD4+ cells expressing each of these cytokines was determined and compared between groups. Unstimulated cells were also stained for CD3 (CD3-PE/Cy5), CD4 (CD4-FITC), CD8 (CD8-BV785), CD45RA (CD45RA-AF700), and CD45RO (CD45RO-PE), and T cell subsets were compared between SLT patients and controls.”

Following on from above, we also assessed the impact of sodium on magnetic bead isolated T cell populations (whole PBMCs, isolated CD4+CD45RA+ cells, isolated CD8+CD45RA+ cells, and isolated CD4+CD45RO+ cells). > 90% purity of cell populations were confirmed with FACS staining.

Purity of cell populations after magnetic bead isolation

We demonstrated in these experiments that sodium affected naïve as well as memory CD4 and CD8 cells, as below. These findings are supported by a recent publication demonstrating sodium chloride affects memory cell populations (Figure 1, Matthias et al, Science translational medicine, 2019, <https://stm.sciencemag.org/content/scitransmed/11/480/eaau0683.full.pdf>). These

results have been added to supplemental figure 6 and the methods updated accordingly.

Sodium effect on naïve and memory CD4+ and CD8+ cells

NaCl effect on naïve and memory CD4+ Cells

NaCl effect on naïve and non naïve CD8+ cells

Sodium responsiveness of PBMCs, naïve and memory CD4+ cells

IL-17 expression plotted in high salt conditions as a ratio to standard conditions; red line drawn at ratio = 1.

Sodium responsiveness IL-17 responses

Given our demonstration that SLT patients don't have differences in naïve and memory cell populations, and that the extracellular ionic environment affects multiple cell subtypes, we used whole PBMCs for polarisation experiments and determined IL-17 expression gated on CD4+ and CD8+ cells. We have not specifically investigated for differences in polarisation of isolated naïve and memory cells between SLT patients and controls for the reasons outlined and have added this to the discussion.

“We demonstrated that SLT patients do not have alterations in naïve and memory cell populations, and that the extracellular ionic environment impacts memory as well as naïve cells confirming recent reports⁶⁴. Hence, we used PBMCs as opposed to isolated cells for IL-17 polarisation experiments. We did not assess all CD4+ subtypes....”

8. STAT1 and STAT3 analysis is done in bulk cell populations raising concerns identical to those above.

Thank you for asking for clarification with regards to the STAT1 and STAT3 analysis. STAT1 and STAT3 phosphorylation were determined in CD4 gated cells after PBMC stimulation. This was undertaken as defects in STAT1 and STAT3 phosphorylation commonly underlie primary IL-17 immunodeficiency resulting in chronic mucocutaneous candidiasis, as outlined in the manuscript. Using stimulated PBMCs is recommended in routine clinical practice for this purpose (see Depner et al,

Journal of Clinical Immunology, 2016,
<https://link.springer.com/article/10.1007/s10875-015-0214-9>). Moreover, flow cytometric analysis of STAT1 and STAT3 gated on CD4+ cells has been shown to correspond well to protein expression, as determined by western blot (see Bitar et al, Journal of Allergy and Clinical Immunology, 2016,
[https://www.jacionline.org/article/S0091-6749\(17\)30915-6/fulltext](https://www.jacionline.org/article/S0091-6749(17)30915-6/fulltext)).

We have now analysed STAT1 and STAT3 phosphorylation in CD4 negative as well as CD4 positive SLT lymphocytes. There was no difference in STAT1 and STAT3 phosphorylation in either lymphocyte population compared to healthy controls. We did not gate on or isolate other T cell subtypes in our analysis but we feel that following this accepted clinical diagnostic protocol is an appropriate method for excluding STAT1 and STAT3 phosphorylation defects as a cause of impaired IL-17 responses in SLT patients.

Supplemental figure 4 has been updated to include these results.

Phosphorylation of STAT3 in CD4- cells after stimulation with IL-6 and IL-21, and phosphorylation of STAT1 in CD4- cells after stimulation with IFN \$\alpha\$ in SLT patients (expressed as ratio of up-regulation in SLT compared to HC; red line drawn at ratio of 1 representing no difference between SLT and HC).

We have also updated the methods (Section: STAT1 and STAT3 phosphorylation assays) including appropriate additional references to clarify why we adopted this approach.

“To investigate for phosphorylation defects in STAT1 and STAT3, PBMCs from SLT patients (n=13) were isolated and cells were stimulated for 15 minutes with IL-6 (100ng/ml), IL-21 (100ng/ml), and IFN α (11500U/ml; R&D Systems, cat#11101-1) according to previously described diagnostic protocols^{70,71}. Cells were fixed immediately (Cytofix Fixation Buffer; BD Biosciences, cat#554655), permeabilised (Phosflow Perm Buffer III; BD Biosciences, cat#558050), and stained for CD4 (CD4-FITC), pSTAT1 (STAT1 pY701 – Alexa647; BD Biosciences, cat#612597) and pSTAT3 (STAT3 pY705 – PE; BD Biosciences, cat#612569). For each subject, the increase in phosphorylation of STAT1 and STAT3 in stimulated compared to unstimulated CD4+ and CD4- cells were determined, and this increase was expressed as a ratio with the increase seen in a healthy control subject in each experiment.”

9. Authors should comment on perceived changes due to magnesium concentration shifts when tables do not suggest differences between patients and controls.

We apologise for any misunderstanding with regards to the magnesium concentrations in SLT patients and disease controls. As demonstrated in supplemental table 4, SLT patients had a significantly lower serum magnesium concentration than the disease control group. We have demonstrated that magnesium promotes IL-17 responses and therefore we propose that reduced extracellular magnesium contributes to infection risk in SLT patients.

Supplemental Table 4: Serum biochemistry in Salt Losing Tubulopathy patients and disease controls.

Values reported are median (IQR), and compared with Mann-Whitney test.

	Salt Losing Tubulopathy	Disease Control	P-value
Na (mmol/l)	140 (138-142)	141 (138-142)	0.54
K (mmol/l)	3.3 (2.9-3.8)	4.3 (4.1-4.5)	<0.0001
Cl (mmol/l)	95 (91-99)	103 (102-104)	<0.0001
HCO ₃ (mmol/l)	29 (26-30)	24 (22-25)	<0.0001
Creatinine (umol/l)	71 (57-88)	81 (62-120)	0.21
cCa (mmol/l)	2.42 (2.37-2.51)	2.39 (2.36-2.43)	0.09
PO ₄ (mmol/l)	1.08 (1.00-1.23)	1.10 (0.87-1.16)	0.42
Mg (mmol/l)	0.70 (0.58-0.79)	0.87 (0.79-0.92)	0.0004

However, we also refer the reviewer to our responses to reviewer 2 with regards to correlating clinical and biochemical parameters to infection score. Whilst we have demonstrated that sodium, potassium, and magnesium (all of which are altered in SLT patients) promote IL-17 responses, blood pressure and renal function were the

only variables to correlate (inversely) with the infection score. This would support a predominant effect of alterations in extracellular sodium, as opposed to other ions, in the development of infection in the SLT cohort, discussed in more detail below.

10. Given aforementioned, overwhelmingly, conclusions seem grandiose and without sufficient data to support them.

We hope the additional data and clarifications above provide further support for our conclusions, although we have tempered these by adding limitations as suggested by the reviewer to the discussion.

Minor comments:

11. Figures appear disjointed.

a. Figure 1A should be of higher quality.

We have provided a clearer representative ^{23}Na -MRI image. It should be noted however that the resolution of ^{23}Na -MRI imaging is less than that of standard hydrogen MRI imaging. This becomes particularly problematic, in our experience, when sodium stores are low (such as in SLT patients) and visualising compartmental sodium in this situation is difficult. During the analysis of ^{23}Na -MRI imaging, we therefore identify regions of interest on the higher resolution hydrogen scans, which are then mapped to the sodium images for determination of sodium concentration at each site.

b. P values are inconsistently presented and are not reflected in figures and scattered around legends.

Thank you for bringing this to our attention. Significance bars have been added to all appropriate figures.

c. Scale bars are not consistent throughout and when compared in subfigures should read the same across.

Thank you for bringing this to our attention. Scale bars within relevant subfigures have been altered and are now consistent.

d. Cell density isn't presented.

e. Are flow cytometry dot plots representative or are they combination of samples? One might assume they are representative dot plots, but they should be explicitly described as such.

f. Dot plots could be labelled in a manner that would make it easier for a reader to understand populations/axes.

g. Dot plots also appear grainy and intensity isn't visible. It should be presented universally throughout main and supplemental figures.

We thank the reviewer for their suggestions with regards to improving the quality and clarity of the FACS data presented. Plots have now been altered such that:

1. cell proportions as percentages are documented within each quadrant (and this has been specified in the figure legends)
2. the legends clearly outline that the plots are representative

3. axis labels have been modified such that they are now larger and in bold and hence plots are easier to understand
4. to improve clarity of the data, large dots have been used for the pseudocolour plots and some pseudocolour plots have been converted to contour plots when populations are clearly defined

Reviewer #2 (Hypertension-immune crosstalk) (Remarks to the Author):

The current studies examine changes in T lymphocyte polarization associated with salt-wasting tubulopathies (SLTs). 47 patients with SLTs are examined. These patients have increased prevalence of infection and allergic phenomena, altered urinary organisms, increased circulating Th2/Th17 lymphocyte ratios, reduced Th17 cell polarization with ex vivo restimulation, elevated levels of innate immunity cytokines, and NK cells showing blunted IFN-gamma production. The analysis is comprehensive. The studies do not elucidate new pathways through which extracellular sodium loss limits Th17 responses, but rather provide an important and compelling confirmatory observation in a unique patient population.

We thank the reviewer for these supportive comments

1. As the epithelial sodium channel (ENaC) has been shown to impact immune cell activation in the setting of hypertonicity, the authors are asked to examine the impact of amiloride on T cell polarization in their patients.

We thank the reviewer for this excellent suggestion. As they allude to, ENaC has been implicated in the mechanism of sodium-mediated inflammation in dendritic cells in mice (Barbaro et al, Cell Reports, 2017). We have now investigated the role of ENaC in T cell responses in human healthy controls, as detailed below.

We first determined lymphocyte expression of the alpha subunit of ENaC by flow cytometry. We demonstrated that ENaC was expressed on multiple lymphocyte subsets (CD4+, CD8+, CD19+, and CD45+CD3-CD56+ cells), more so on naïve (CD45RA+) and CD8+ cells compared to memory (CD45RA-) and CD4+ cells respectively.

Lymphocyte ENaC expression

Data represent expression on CD4, CD8, and CD45RA cells determined in 7 healthy controls. CD4+ 30% (23-43), CD8+ 41% (35-52), p=0.06. CD45RA- 30% (21-38), CD45RA+ 53% (48-64), p=0.03. Compared with Wilcoxon test.

We then undertook Th17 polarisation experiments, as described in the manuscript, in media supplemented with 40mM NaCl with and without sodium channel and transporter inhibitors (amiloride, hydrochlorothiazide, furosemide, and the NCX inhibitor KB-R7943) to determine whether sodium driven inflammation could be abrogated. Amiloride reduced supernatant IL-17 concentration in 11 healthy controls. Consistent with the increased expression of ENaC on CD8+ cells, this was associated with a reduction in Tc17 polarisation, which was also seen with the addition of furosemide (below). We demonstrated that amiloride had a direct effect on reducing IL-17 responses in isolated naïve CD4+ and CD8+ cells. Moreover, we showed that sodium alters the pattern of calcium flux during T cell activation and this effect is reversed with amiloride.

Effect of amiloride (and other sodium transport inhibitors) on IL-17 responses

The effect of amiloride 20uM (n=11), furosemide 20uM (n=13), Hydrochlorothiazide 20uM (n=11), and KB-R7943 200nM (n=8) on sodium driven IL-17 responses (healthy controls). Variables (supernatant IL-17 concentration, Th17 and Tc17 polarisation) in the presence of inhibitor are expressed as a ratio to culture without the inhibitor present. Red lines drawn at ratio = 1 represent no difference between conditions. Compared with Wilcoxon test.

Effect of amiloride on IL-17 expression in isolated naïve CD4+ and CD8+ cells
 Isolated naïve CD4 and CD8 population purity was confirmed prior to 7-day stimulation in the presence of Th17 polarising cytokines. Technical replicates (x2 for each condition) within an individual are shown.

Effect of sodium and amiloride on calcium flux after T cell activation

Calcium flux was measured with the ratiometric Indo-1 calcium dye. T cells were activated by anti-CD3

Calcium flux in CD4+ cells after T cell activation in the presence of NaCl.

Amiloride effect on calcium flux during T cell activation

These preliminary data support the role of amiloride-sensitive sodium transport mechanisms in human sodium driven IL-17 responses, through a mechanism involving altered calcium flux. It is important to note we have not included SLT patients with inactivating ENaC mutations (pseudohypoaldosteronism type 1) – these patients salt waste but have a different biochemical phenotype to patients with loop or distal convoluted tubule dysfunction with hyperkalaemic acidosis and no hypomagnesaemia. Whilst we have shown ENaC mediates sodium driven IL-17 responses, we do not anticipate that primary alterations in sodium transporters underlie reduced IL-17 responses in our SLT cohort and this is supported by their normal calcium flux during T cell activation, the lack of an effect of sodium chloride cotransporter inhibition on IL-17 responses (in the case of GS), and the inclusion of patients with multiple ion channel/transporter mutations within the cohort.

The results pertaining to ENaC function in T cells outlined above form part of a separate piece of ongoing work investigating the mechanisms of sodium driven inflammation and therefore we have not included them in this submission. As highlighted in the manuscript, however, we outline that we do not think that defective sodium transport mechanisms on immune cells are responsible for the immune phenotype in SLT patients (see Results section 7: STAT1 and STAT3 phosphorylation are normal in SLT patients and reduced IL-17 responses are not due to altered ion channel or transporter activity on immune cells) and we have also expanded the discussion to make this clear:

“Moreover, in mice, impaired SOCE due to deletion of ORAI1 or STIM1 impairs Th17 polarisation^{58,59}, and sodium has been shown to exert its proinflammatory effect in dendritic cells through epithelial sodium channel (ENaC) flux altering calcium signaling⁶⁰. In SLT, we demonstrate that calcium flux is unaffected after T cell activation and patients with inactivating ENaC mutations (pseudohypoaldosteronism type 1) were not included in the cohort. Whilst we demonstrate the presence of other sodium transport mechanisms on diverse immune cells, this unaltered calcium flux, alongside the lack of an effect of sodium chloride cotransporter inhibition on Th17 polarisation, and the inclusion of patients with multiple ion

channel/transporter mutations in our cohort make reduced function of channel/transporter activity on immune cells an unlikely reason for altered immunity.”

2. The authors find that kidney function inversely correlates with infection scores. The authors are asked to examine the relationship between blood pressure and infection as it seems plausible that sodium loss would lead both to lower blood pressures and higher rates of infection.

We are very grateful to the reviewer for suggesting this. We have now analysed the relationship between blood pressure (as a surrogate for sodium loss) and infection score. As the reviewer anticipates, there is a close inverse correlation between blood pressure (in particular diastolic blood pressure) and IL-17 related infection score. This would support a predominant effect of salt depletion and reduced extracellular sodium on infection risk over changes in other extracellular ions. This may explain the differences in infection rate between SLT types, as discussed in more detail in response to reviewer 3’s comments below. As suggested by reviewer 3, figures depicting differences in infection rate and serum creatinine have been added to the main section of the manuscript (Figure 7F-G).

The results (section 8) have been updated to reflect this blood pressure analysis:

“We then correlated serum biochemical and clinical parameters in SLT with their IL-17 related infection score. Impaired renal function as demonstrated by increased serum creatinine leads to reduced salt loss and blood pressure may act as a surrogate for sodium stores^{33,34}. In support of a predominant effect of salt depletion and reduced extracellular sodium on immunity, blood pressure and serum creatinine inversely correlated most closely with infection score (diastolic blood pressure Pearson r -0.37 (95% CI -0.60, -0.09), p=0.013; mean arterial pressure Pearson r -0.32 (95% CI -0.56, -0.03), p=0.03; serum creatinine Pearson r -0.27 (95% CI -0.52, 0.02), p=0.06), whilst there was no correlation of infection score with serum magnesium or potassium (Supplemental figure 7A-B). Moreover, aligned with their differences in renal function, there was a difference in infection score between SLT types grouped according to biochemical phenotype, with infection score highest in patients with GS and EAST syndrome (distal tubule dysfunction and most preserved renal function), and lowest in patients with BS types 1,2 and 4 (loop dysfunction and most impaired renal function) (Figure 7F-G).”

We have also updated supplemental figure 7 with these results:

Supplemental Figure 7: IL-17 related infection score according to serum biochemical parameters and disease subtypes in Salt Losing Tubulopathy (SLT) patients

A. Correlation of serum biochemical parameters to IL-17 related infection score in SLT patients; analysed with Pearson correlation.

B. Mean arterial pressure, diastolic blood pressure, and serum creatinine plotted against IL-17 related infection score in SLT patients.

ns – not significant ($p > 0.05$), * $p \leq 0.05$, ** $p \leq 0.01$, *** $p \leq 0.001$, **** $p \leq 0.0001$

A.

	Pearson r	95% Confidence Interval	R squared	P value
Infection Score vs. Na (mmol/l)	-0.1654	-0.4321 to 0.1278	0.02737	0.2664
Infection Score vs. K (mmol/l)	-0.09249	-0.3728 to 0.2033	0.008554	0.541
Infection Score vs. Cl (mmol/l)	-0.2305	-0.4908 to 0.06760	0.05313	0.1277
Infection Score vs. HCO ₃ (mmol/l)	0.0489	-0.2449 to 0.3345	0.002391	0.7469
Infection Score vs. Creatinine (umol/l)	-0.2718	-0.5185 to 0.01666	0.07388	0.0646
Infection Score vs. cCa (mmol/l)	-0.1446	-0.4203 to 0.1556	0.0209	0.3434
Infection Score vs. PO ₄ (mmol/l)	-0.239	-0.4950 to 0.05510	0.05713	0.1096
Infection Score vs. Mg (mmol/l)	-0.04587	-0.3450 to 0.2617	0.002104	0.773
Infection Score vs. Mean arterial pressure (mmHg)	-0.3233	-0.5634 to -0.03290	0.1045	0.0303
Infection Score vs. Systolic blood pressure (mmHg)	-0.1988	-0.4652 to 0.1006	0.03951	0.1905
Infection Score vs. Diastolic blood pressure (mmHg)	-0.3695	-0.5982 to -0.08526	0.1366	0.0125

B.

We have updated the methods (section: Salt losing nephropathy and control cohorts) accordingly:

“We also compared infection scores between SLT subtypes and we correlated infection score to serum biochemical and clinical parameters in SLT patients.”

We have also highlighted this analysis in the discussion:

“We also provide evidence for the first time that sodium depletion affects immunity and leads to IL-17 mediated immunodeficiency. Moreover, within the SLT cohort, blood pressure and serum creatinine inversely correlated to infection score, and hence we propose that extracellular sodium as opposed to alterations in other ions is the main determinant of infection risk. This requires confirmation with direct measurement of sodium stores in a larger number of SLT patients as part of future study.”

3. Last, as interstitial sodium is thought to impact dendritic cell priming, the authors are asked to provide data if available regarding the capacity of the patients’ dendritic cells to activate T lymphocytes.

We thank the reviewer for this interesting suggestion. As alluded to above, dendritic cells are affected by interstitial sodium, in mice through a mechanism involving ENaC (Barboro et al, Cell reports, 2017). As we have described, we have data to suggest that ENaC is also directly implicated in sodium driven IL-17 responses in T cells. Importantly, we have not included SLT patients with inherited defects in ENaC (Pseudohypoaldosteronism Type 1) in this analysis. The sodium transport defects represented in our SLT cohort are not known to be directly implicated in dendritic cell function.

We feel a primary defect in dendritic cells is unlikely for the following reasons:

1. The clinical phenotype of SLT patients is not consistent with primary dendritic cell immunodeficiency

Primary deficiencies in dendritic cells may be due to mutations in GATA2 (Bigley et al, J Exp Med, 2011) and IRF8 (Hambleton et al, NEJM, 2011). The major infective manifestations in these diseases are recurrent viral and mycobacterial infections, which were not seen in the SLT cohort. The clinical phenotype of SLT patients is much more consistent with primary immunodeficiencies that alter IL-17 responses, as highlighted in Supplemental table 6.

2. The specific defect in Th17 cells as opposed to a more generalised defect in T cell function is not consistent with impaired antigen presentation by dendritic cells

Dendritic cells promote T cell responses through the presentation of antigen on MHC class II and the production of pro-inflammatory cytokines. Defects in dendritic cell function would be expected to impact T cells more generally than the specific Th17 defect, which is encountered in the SLT cohort (see Figure 3).

3. Whole blood assays to investigate innate cytokine responses are not consistent with a defect in dendritic cells

Dendritic cells recognise antigen via toll like receptors and produce pro-inflammatory cytokines such as IL-12, IFN γ , TNF α , IL-6, and IL-10. In dendritic cell deficiency (e.g. GATA2 deficiency), these cytokine responses are defective in whole blood stimulation assays (see figure 1F-H in Hambleton et al, NEJM, 2011 – these are the same cytokine stimulation assays as were used in our cohort). There was no defect in these cytokine responses in the SLT patients – indeed responses in many of these cytokines were in fact increased in SLT patients (see results Section 10: SLT patients have increased innate immune responses).

Given the above, we did not perform further investigation of dendritic cell function in SLT patients. We have highlighted this in the discussion, and outline the reasons for not doing so as we have above:

“We did not assess all CD4+ subtypes and any effect on regulatory T cells or T follicular helper cells, both of which may be affected by sodium^{14,15,64}, in SLT patients was unexplored. Similarly, sodium has been shown to impact dendritic cell function⁶⁰, but the clinical phenotype of SLT patients (absent recurrent viral and mycobacterial infections) and the specific defect in IL-17 responses were inconsistent with primary defects in dendritic cells^{65,66}. Moreover, innate cytokine responses in whole blood assays were unaffected and hence further functional assessment of dendritic cells was not undertaken.”

Reviewer #3 (Salt metabolism) (Remarks to the Author):

General Comments: Evans and colleagues show that immune cells from patients with mutations of electrolyte transporters display reduced Th17 polarization, which can be reversed in vitro by increasing the NaCl concentration in the cell culture medium by 40 mmol/L. Because this immune cell phenotype is associated with a higher incidence of infections, and because some of the patients show lower skin Na content, the authors conclude that chronic (renal) salt wasting associates with immunodeficiency due to an altered extracellular ionic environment that impacts T cell polarization. To my opinion, this is an important clinician-scientist study which capitalizes on a relevant and carefully characterized cohort of patients with electrolyte transporter mutations. My major criticism is that the patho-physiological connection between renal effects, skin microenvironmental effects, and immune cell function that the authors construct is not (yet) substantiated by the data.

We thank the reviewer for these supportive comments. We have provided further data and clarifications to support our conclusions in response to the specific comments below.

Specific Comments:

1. General Approach – Physiology:

(i) The authors construct the view that chronic salt loading leads to “non-osmotic sodium storage in muscles and the skin, resulting in interstitial sodium concentrations at these sites in excess of those in plasma,” and that this sodium storage will activate immune cells. This construct is not helpful. In their in-vitro experiments, the authors increase the Na concentration in the cell culture medium by 40 mmol/L and thereby induce an osmotic stress reaction with osmotically active sodium. Indeed, many of the described hypertonicity-driven immune cell responses take place in-vivo in the skin at tissue (Na+K) / tissue water ratios of approximately 190 mmol/L, suggesting the presence of a hypertonic environment. Thus, the authors may want to prevent the traditional term “non-osmotic sodium storage”, which derives from an era where investigators believed that the sodium concentrations in blood and in peripheral extracellular fluids are not much different, and that relevant osmotic gradient formation only exists in the kidney. This view is most likely not correct; the skin might alternatively be viewed as an osmotic gradient-generating, kidney-like counter current system (Pflugers Arch. 2015 Mar;467(3):551-8).

We thank the reviewer for this suggestion, and for their fascinating insights in to the potential of the skin containing an osmotic gradient-generating system akin to the counter-current system in the kidney. We have removed the concept of the skin microenvironment being ‘non-osmotic’ as the reviewer suggests in both the introduction and the discussion. We have also added the above reference and highlighted the potential of the skin to generate osmotic/sodium gradients with further discussion around this point as outlined in response to the comments below.

Introduction:

“Chronic salt loading leads to sodium storage in muscles and the skin, resulting in interstitial sodium concentrations at these sites in excess of those in plasma⁸.

Sodium here may be cleared.....”

Discussion:

“Novel findings from long-term sodium balance studies have demonstrated a third compartment of sodium stored at interstitial sites in the muscle and skin. It is proposed that the skin contains as an osmotic gradient-generating system akin to the countercurrent mechanism in the kidney³⁸. At both sites increased sodium concentrations prevent excess water loss but may also provide protection from infection^{11,18}. Whilst the sodium transport mechanisms in the kidney are well

described, ion channels and transporters involved in skin sodium transport are largely unknown. Visualization and measurement of skin sodium.....”

(ii) In line with the above comments, it is unclear to me why the authors conclude from the data they present that the reduction in skin Na concentration in patients with mutations of electrolyte transporters are secondary to renal salt wasting: in Supplemental Table 2, the authors report normal serum Na concentrations in their patients, while in Figure 1 they report a reduction in skin Na concentration. My conclusion is that the patient’s kidneys have achieved constancy in the plasma Na microenvironment, while the interstitial Na concentration specifically in the skin (not in muscle) is reduced, suggesting an extrarenal problem (keratinocyte transport?). My understanding after having had a quick look at the tissue-specific expression of reported mutated genes in the patients (Supplemental Table 1) at <http://biogps.org/#goto=welcome> is that at least the described transporter mutations in patients with Gitelman and EAST syndromes are not kidney specific. I conclude that MRI analysis of potential differences in the skin sodium storage phenotype between patients with Bartter syndrome and patients with Gitelman or EAST syndromes might provide with important additional insights that could further substantiate the authors hypotheses on the importance of the ionic microenvironment for immunological host defense.

We thank the reviewer for this interesting suggestion, and raising the possibility that skin sodium concentrations are reduced in some SLT patients on account of defective keratinocyte sodium transport, as opposed to/in addition to renal salt wasting. As the reviewer alludes to, we included SLT patients with defects in a number of ion transporters, many of which are not renal specific. NKCC2 (encoded by SLC12A1) and ROMK (encoded by KCNJ1) are the most renal specific of these transporters, and result in Bartter syndromes 1 and 2 respectively.

The ²³Na-MRI scans reported in the original Figure 1 were undertaken in patients with Gitelman Syndrome. We have now also undertaken ²³Na-MRI imaging in 2 patients with renal specific transporter defects resulting in BS1, one with more impaired renal function than the other. Reduced skin Na was demonstrated in one of the BS1 patients as it was in GS and hence we propose that the predominant reason for reduced skin sodium concentrations in SLT is due to renal salt wasting as opposed to primary defect in skin sodium transport.

Skin sodium concentration in SLT patients (GS = red dots; BS1 (renal specific) = blue dots) and healthy controls

We have also analysed the relationship between serum creatinine (as a marker of renal function) and skin sodium concentrations to try and understand the differences between skin sodium storage within the SLT patients. Albeit in only 6 SLT patients, skin Na increased according to serum creatinine supporting a protective effect of impaired renal function on skin sodium stores within the SLT patients.

Relationship between skin Na concentration and serum creatinine in SLT patients (GS = red dots; BS1 (renal specific) = blue dots)

Figure 1 and the discussion have been updated to reflect this additional analysis.

Discussion:

“While SLT patients waste sodium, how this affects sodium stores or if this impacts on total body sodium balance in the long term is **unexplored**. Moreover, if the **defective ion transport mechanisms in SLT patients directly impact skin sodium**

transport is unknown. Using ^{23}Na -MRI, we demonstrated that sodium stores are reduced in SLT patients compared to controls. This reduction was seen in SLT patients with defects in both renal specific (BS1) and systemic (GS) transporters suggesting the predominant reason for reduced skin sodium is renal sodium wasting as opposed to a primary defect in keratinocyte sodium transport in the skin. This alteration in interstitial sodium combined with reduced serum potassium and magnesium concentrations mean immune cells in SLT patients are exposed to a unique combination of altered extracellular ionic cues, which we propose impacts immune function.”

2. General Approach – Immunology:

Similarly, patients with Bartter, Gitelman, or EAST syndromes may show different levels of immune cell activation, because their immune cells may express different levels of mutated transporters in their cell membranes (<http://biogps.org/#goto=welcome>). This may be relevant for host defense mechanisms, as well as for renal and extrarenal maintenance of local salt and water homeostasis.

We thank the reviewer for raising the issue of differences in immune cell activation and therefore infection risk dependent on the transporter affected. In addition to the comments below, we also refer the reviewer to our comments in reply to reviewer 2 above with regards to our investigation of ENaC and T cell reactivity, and our analysis of the relationship between clinical and biochemical parameters and infection score.

We included SLT patients with defects in 6 different ion transport mechanisms (resulting in 4 different types of BS, GS, and EAST syndrome respectively). As the reviewer alludes to some of these transporters may be expressed on immune cells. We have demonstrated this to be the case for the sodium chloride cotransporter (defective in GS) as outlined in supplemental figure 4 in the manuscript, as we too had considered primary dysfunction of ion transporters on immune cells to be a potential mechanism for altered IL-17 responses. We have now assessed IL-17 immunity and infection risk (as determined by infection score) according to defective transporter as suggested by the reviewer. We found no significant differences in either the infection score or in Th17 polarisation according to the individual transporter affected.

Infection score and Th17 polarisation according to SLT type

In addition to this data, we feel primary dysfunction of ion transporters on immune cells is an unlikely cause of impaired immunity due to:

1. a lack of an IL-17 abrogating effect with transporter inhibition

As outlined in figure 5, the addition of hydrochlorothiazide (inhibitor of NCC – defective in GS) had no impact on IL-17 responses.

Fig 5C. Th17 and Tc17 polarisation, and supernatant IL-17 concentration in experiments (n=11) with the addition of hydrochlorothiazide (HCT) 20uM to culture conditions (expressed as ratio to readout in standard culture conditions; red line drawn at ratio of 1 representing no difference in readouts with the addition of HCT).

2. SLT patients have normal calcium flux during T cell activation

As we have described in our responses to reviewer 2, and as has been previously published (Barboro, Cell Reports 2017), increased extracellular sodium mediates inflammation by altering calcium flux in immune cells. Inhibition of or defects in sodium transporters/channels (e.g ENaC) have opposing calcium altering effects. As

outlined in Figure 5, calcium flux after T cell activation was no different in SLT patients compared to healthy controls.

Fig5D: Area under the calcium flux curve (as determined using the ratiometric calcium dye Indo-1 gated on CD4+ cells) after T cell activation with anti-CD3: HC 275.9AU (265.9-279.9); and Gitelman Syndrome (GS) 277.8AU (257.5-280.5), $p=0.94$. Compared with Mann-Whitney test.

Fig 5E. Representative calcium flux curves in CD4+ cells after T cell activation in a healthy control and 2 patients with GS.

The results (section 7) and supplemental figure 4 have been updated to include this additional analysis:

“...We investigated NCC expression on immune cells and assessed whether its inhibition led to reduced IL-17 responses. NCC was expressed on diverse immune cell subtypes but NCC inhibition with hydrochlorothiazide (HCT; 20uM) had no effect on *in vitro* IL-17 responses in cells from HCs (**Figure 5C; Supplemental figure 4D-F**). SLT patients had a normal pattern of calcium flux after T cell activation (**Figure 5D-E**). Furthermore, infection score and IL-17 responses were not different according to the ion transporter affected (**Supplemental figure 4G**). Given these data, reduced IL-17 responses in SLT are not likely to be due to primary perturbations of NCC function or secondary dysregulation of intracellular calcium signaling in immune cells.

Whilst the above data support no defect in ion transporters on immune cells, to confirm this patch clamping SLT lymphocytes and measurement of ionic flux would be required, and this was not undertaken. The discussion has been updated to clarify our reasoning behind why we feel primary defects in ion transporters on immune cells is an unlikely reason for altered immunity, and we also add the lack of confirmatory measurement of lymphocyte sodium flux as a limitation.

Discussion:

“Moreover, in mice, impaired SOCE due to deletion of ORAI1 or STIM1 impairs Th17 polarisation^{59,60}, and sodium has been shown to exert its proinflammatory effect in dendritic cells through epithelial sodium channel (ENaC) flux altering calcium signaling⁶¹. In SLT, we demonstrate that calcium flux is unaffected after T cell activation and patients with inactivating ENaC mutations (pseudohypoaldosteronism type 1) were not included in the cohort. Whilst we demonstrate the presence of other sodium transport mechanisms on diverse immune cells, this unaltered calcium flux, alongside the lack of an effect of sodium chloride cotransporter inhibition on Th17 polarisation, and the inclusion of patients with multiple ion channel/transporter mutations in our cohort make reduced function of channel/transporter activity on immune cells an unlikely reason for altered immunity.”

Discussion:

“We measured calcium flux during T cell activation but were unable to measure sodium flux in patients or controls and hence were unable to confirm that SLT

patients don't have a primary defect in ion transport mechanisms on immune cells.

SLT patients.....”

3. Data presentation – Fig. 1:

The authors show a convincing reduction in skin Na content in “SLT” patients.

However, the grouping of “SLT” patients is not convincing, because it is not clear whether “SLT” refers to patients with Bartter, Gitelman, or EAST syndrome (n=4). I have 3 specific questions:

1. In patients with Bartter syndrome, one would assume a quite kidney-specific effect of the SLC12A1 mutation, and not much effect in keratinocytes (see comment 1). Is skin Na content in patients with Bartter syndrome different from controls, and/or different from patients with EAST or Gitelman syndrome?

We thank the reviewer for this suggestion. As outlined in response to point 2, reduced sodium stores were demonstrated in patients with both renal specific (BS type 1) and systemic (GS) transporter defects.

2. In patients with Gitelman syndrome, one would assume a less kidney-specific effect, and an additional effect of the SLC12A3 mutation on keratinocyte transport, which might be mediated by dendritic cell function (see comments 1 and 2). Is skin Na content in patients with Gitelman syndrome different from controls, and/or different from patients with Bartter or EAST syndrome?

Skin sodium concentrations were reduced in GS compared to controls. However, reduced skin Na was also encountered in renal specific BS type 1, as described.

3. In patients with EAST syndrome, one would assume that extrarenal effects of the mutated transporter, including keratinocyte transport, are at least as relevant as renal-tubular effects (see comments 1 and 2). Is skin Na content in patients with EAST syndrome different from controls, and/or different from patients with Bartter or Gitelman syndrome?

We would anticipate that skin sodium storage in EAST syndrome is equivalent to GS, given both syndromes result in the same tubular phenotype. Due to expression of KCNJ10 (mutated in EAST syndrome) in the central nervous system, these patients have severe ataxia in addition to the tubular features which means ²³Na-MRI imaging is logistically challenging and we have not been able to undertake scans in either of the EAST patients in our cohort.

4. Data presentation – Fig. 2, IL-17-related infection scores.

Again, the data presentation suggesting that the observed changes are secondary to renal tubular transporter problems, may be misleading. Given the fact that KCNJ1 mutations in patients with EAST syndrome are not really kidney-specific, that SLC12A3 mutations in patients with Gitelman syndrome may also be pretty DC-specific, while SLC12A1 mutations in patients with Bartter syndrome may be much more kidney-specific: are there differences in Th17-related infection scores between patients with EAST, Gitelman, or Bartter syndromes? Online Supplemental Figure 6C suggests that patients with EAST and Gitelman syndromes indeed have more

infections. Why is this finding buried in the Online Supplement? Could it be that these patients have higher infection scores compared to patients with Bartter Syndrome, because (i) skin Na contents are different? (ii) because KCNJ1 mutations (EAST) or SLC12A3 mutations (Gitelman) in immune cells reduce their ability to ward-off infections, compared to patients with Bartter syndrome, in whom SLC12A1 is not relevant for modulating immune cell function?

We thank the reviewer for these interesting ideas with regards to differences in infection risk within the SLT cohort, and potential reasons for these.

As outlined above, we found no differences in infection score or IL-17 responses according to the individual transporter defective, and we feel that transporter dysfunction on immune cells is an unlikely mechanism for impaired immunity for the reasons described in reply to comment 2. We accept as a limitation the lack of direct sodium measurement in immune cells as discussed.

We do agree with the reviewer, however, that there are differences within the SLT cohort with regards to infection risk, which may relate to their biochemical phenotypes. BS types 1,2, and 4 have pure loop dysfunction, GS and EAST have pure distal tubule dysfunction, whereas BS type 3 patients have mixed loop/distal tubule dysfunction on account of expression of CLC-Kb at both sites. This leads to differing biochemical phenotypes between the groups, as outlined in supplementary table 3.

Supplemental Table 3: Serum biochemistry in the Salt Losing Tubulopathy (SLT) Cohort.

Values reported are mean (standard deviation). Variables are compared between BS types 1,2, and 4 ('antenatal BS/loop phenotype'), BS type 3 ('classical BS/mixed loop and distal tubule phenotype'), and GS/EAST syndrome ('distal tubule phenotype') with a one-way analysis of variance.

	Whole SLT cohort n=47	Bartter syndrome types 1,2, and 4 n=9	Bartter syndrome type 3 n=14	Gitelman and EAST syndrome n=24	P-value
Na (mmol/l)	140 (3)	141 (4)	140 (3)	140 (3)	0.68
K (mmol/l)	3.3 (0.6)	3.7 (0.6)	2.9 (0.6)	3.4 (0.5)	0.0045
Cl (mmol/l)	95 (4)	99 (2)	91 (4)	96 (3)	<0.0001
HCO3 (mmol/l)	28 (4)	25 (3)	30 (4)	28 (3)	0.014
Creatinine (umol/l)	94 (79)	166 (140)	103 (63)	62 (14)	0.0005
cCa (mmol/l)	2.43 (0.13)	2.41 (0.12)	2.41 (0.14)	2.45 (0.12)	0.59
PO4 (mmol/l)	1.11 (0.20)	1.19 (0.25)	1.13 (0.21)	1.06 (0.16)	0.20
Mg (mmol/l)	0.70 (0.17)	0.79 (0.18)	0.75 (0.16)	0.64 (0.16)	0.05

Notably there are differences in excretory renal function between the groups with BS 1,2 and 4 having more impaired renal function than GS and EAST. This has been emphasised in the manuscript by adding the below data to figure 7.

Serum creatinine according to SLT type: BS types 1,2, and 4 = 108 μ mol/l (81-240), BS type 3 = 77 μ mol/l (56-162), GS and EAST 66.5 μ mol/l (52-73), p=0.0005. Groups compared with One-way analysis of variance and multiple comparison testing

We found a difference in infection scores when SLT patients were compared between these groups (as opposed to no difference when classifying by individual transporter as above). With these 'biochemical' groupings there was increased infection in GS and EAST compared to BS 1,2, and 4. The previously supplemental figure detailing this has been moved to the main manuscript (figure 7) as the reviewer suggests.

IL-17 related infection score according to SLT type: BS types 1,2, and 4 = 2.0 \pm 2.1; BS type 3 = 4.6 \pm 3.3; GS and EAST = 5.7 \pm 4.5, p=0.05. Groups compared with One-way analysis of variance and multiple comparison testing.

We propose that these differences are related to differences in the biochemical profiles of the groups, in particular differences in extracellular sodium concentration. As outlined in our response to reviewer 2, we correlated clinical and biochemical parameters within the SLT cohort to infection score to try and understand these differences in infection rate. We demonstrate that blood pressure (as a surrogate for sodium storage) and serum creatinine (as a marker of impaired renal function and therefore a cause of increased sodium retention) inversely correlated most closely with infection score, whereas other biochemical parameters that were different between the groups (e.g. potassium and magnesium) did not (see updated supplemental figure 7 below). Indeed, in those that underwent ^{23}Na -MRI imaging (HC and SLT patients), infection score reduced as measured skin sodium storage increased.

Relationship between measured skin sodium concentrations and IL-17 infection score in healthy controls (black dots) and SLT patients (BS1 = blue dots; GS = red dots)

Spearman r correlation coefficient -0.61 (p=0.068)

We would like to support this proposal of extracellular sodium being the predominant driver of infection risk with direct measurement of sodium stores (and correlate these with immunity) in a larger number of SLT patients in future work, and have updated the discussion accordingly.

Updated supplementary figure 7:

Supplemental Figure 7: IL-17 related infection score according to serum biochemical parameters and disease subtypes in Salt Losing Tubulopathy (SLT) patients

A. Correlation of serum biochemical parameters to IL-17 related infection score in SLT patients; analysed with Pearson correlation.

B. Mean arterial pressure, diastolic blood pressure, and serum creatinine plotted against IL-17 related infection score in SLT patients.

ns – not significant ($p > 0.05$), * $p \leq 0.05$, ** $p \leq 0.01$, *** $p \leq 0.001$, **** $p \leq 0.0001$

A.

	Pearson r	95% Confidence Interval	R squared	P value
Infection Score vs. Na (mmol/l)	-0.1654	-0.4321 to 0.1278	0.02737	0.2664
Infection Score vs. K (mmol/l)	-0.09249	-0.3728 to 0.2033	0.008554	0.541
Infection Score vs. Cl (mmol/l)	-0.2305	-0.4908 to 0.06760	0.05313	0.1277
Infection Score vs. HCO ₃ (mmol/l)	0.0489	-0.2449 to 0.3345	0.002391	0.7469
Infection Score vs. Creatinine (umol/l)	-0.2718	-0.5185 to 0.01666	0.07388	0.0646
Infection Score vs. cCa (mmol/l)	-0.1446	-0.4203 to 0.1556	0.0209	0.3434

Infection Score vs. PO4 (mmol/l)	-0.239	-0.4950 to 0.05510	0.05713	0.1096
Infection Score vs. Mg (mmol/l)	-0.04587	-0.3450 to 0.2617	0.002104	0.773
Infection Score vs. Mean arterial pressure (mmHg)	-0.3233	-0.5634 to -0.03290	0.1045	0.0303
Infection Score vs. Systolic blood pressure (mmHg)	-0.1988	-0.4652 to 0.1006	0.03951	0.1905
Infection Score vs. Diastolic blood pressure (mmHg)	-0.3695	-0.5982 to -0.08526	0.1366	0.0125

The results (section 8) have been updated to reflect this analysis:

“We then correlated serum biochemical and clinical parameters in SLT with their IL-17 related infection score. Impaired renal function as demonstrated by increased serum creatinine leads to reduced salt loss and blood pressure may act as a surrogate for sodium stores^{33,34}. In support of a predominant effect of salt depletion and reduced extracellular sodium on immunity, blood pressure and serum creatinine inversely correlated most closely with infection score (diastolic blood pressure Pearson r -0.37 (95% CI -0.60, -0.09), p=0.013; mean arterial pressure Pearson r -0.32 (95% CI -0.56, -0.03), p=0.03; serum creatinine Pearson r -0.27 (95% CI -0.52, 0.02), p=0.06), whilst there was no correlation of infection score with serum magnesium or potassium (Supplemental figure 7A-B). Moreover, aligned with their differences in renal function, there was a difference in infection score between SLT types grouped according to biochemical phenotype, with infection score highest in patients with GS and EAST syndrome (distal tubule dysfunction and most preserved renal function), and lowest in patients with BS types 1,2 and 4 (loop dysfunction and most impaired renal function) (Figure 7F-G).”

Discussion:

“We also provide evidence for the first time that sodium depletion affects immunity and leads to IL-17 mediated immunodeficiency. Moreover, within the SLT cohort, blood pressure and serum creatinine inversely correlated to infection score, and

hence we propose that extracellular sodium as opposed to alterations in other ions is the main determinant of infection risk. This requires confirmation with direct measurement of sodium stores in a larger number of SLT patients as part of future study.”

5. Data presentation – Fig. 4 and Fig. 8, reduction in Th17 polarization patterns; and recovery by increasing the salt concentration in the cell culture medium. In line with comment 4, are there differences in Th17-dependent polarization patterns between cells from patients with EAST, Gitelman, or Bartter syndromes? Does NaCl-driven recovery depend on differences in transporter mutation?

We thank the reviewer for suggesting this analysis. Th17 polarisation and salt responsiveness (i.e. recovery) were no different when analysed according either to individual transporter defective or when analysed according to nephron location of defective transporter ('biochemical grouping' as above).

Th17 polarisation and Th17 salt responsiveness (i.e. Th17 polarisation in high salt media expressed as ratio to standard media; red line drawn at ratio = 1) according to defective transporter

Th17 polarisation according to SLT Type

Salt responsiveness (Th17) according to SLT type

Th17 polarisation and Th17 salt responsiveness (i.e. Th17 polarisation in high salt media expressed as ratio to standard media; red line drawn at ratio = 1) according to location of defective transporter ('biochemical SLT type')

Th17 polarisation according to SLT Type

Salt responsiveness according to SLT type

These findings have been added to supplemental figure 4G and supplemental 8D have been updated to include these results.

The results (sections 7 and 9) have also been updated:

Section 7:

“SLT patients had a normal pattern of calcium flux after T cell activation (**Figure 5D-E**). Furthermore, infection score and IL-17 responses were not different according to the ion transporter affected (**Supplemental figure 4G**).”

Section 9:

“Salt responsiveness of *in vitro* IL-17 responses in SLT patients was no different to controls, and there were no differences in salt responsiveness according to SLT type (**Supplemental figure 8B-D**). These data demonstrate that salt supplementation ameliorates IL-17 response defects in SLT *in vitro*, but whether this is feasible and effective clinically *in vivo* is unknown.”

6. Discussion: Data interpretation – “role of channel/transporter activity on immune cells an unlikely reason for altered immunity.”

Given the fact that patients with Gitelman syndrome may express large amounts of mutated SLC12A3 specifically in dendritic cells (<http://biogps.org/#goto=genereport&id=6559>), I wonder whether the authors may have overlooked the role of DCs in this context. DC residing in the low-salt interstitium in patients with Gitelman syndrome may reduce their co-stimulatory signals, resulting in a generally reduced IL17 response in T cells.

We thank the reviewer for this suggestion. We refer them to our response to reviewer 2 (last comment) with regards to dendritic cell function.

7. Discussion:

Data interpretation – the idea that these exciting environmental changes in the skin reported here could be tackled by increasing daily salt consumption may be naïve. We meanwhile know that extrarenal, local activation of salt retaining mechanisms are key for the induction of Na storage, and that this Na storage does not readily respond to changes in dietary salt intake. From a balance-point-of-view, this is not very surprising. A human keratinocyte is exposed to an estimated 5 L/min x 7200 min x 9 g/L = 64.8 kg salt per day, which is offered to the cell via the blood circulation. Do the authors really believe that a 0.006 – 0.020 kg/d variation in salt intake will significantly change this local salt exposure?

We thank the reviewer for this insight in to dietary changes and skin sodium exposure. We accept that correcting renal salt loss in SLT patients is challenging with oral supplementation / salt ad libitum in their diet, and we have updated the discussion accordingly. We do, however, have a number of SLT patients who regularly consume > 20g salt /day. Whilst we do not know the impact of maximising salt intake on SLT sodium stores, small changes in extracellular sodium coupled with

changes in other electrolytes we have shown to impact immunity (i.e. potassium and magnesium) may impact infection risk, and therefore warrants further study.

The abstract conclusion and main text discussion have been updated.

Abstract

“Conclusion

Chronic salt wasting in inherited SLT is associated with immunodeficiency due to impaired IL-17 responses. This is due to an altered extracellular ionic environment impacting T cell polarisation. Whether **better correction of extracellular ions** can rescue the immunophenotype *in vivo* in SLT patients is unknown.”

Discussion:

“Given this proposal of ionic changes altering immunity in SLT, we lastly tested whether manipulation of the extracellular ionic environment *in vitro* could be used to augment SLT IL-17 responses, as we had shown in controls. We demonstrated that the intracellular pathway that mediates sodium driven Th17 polarisation is intact in SLT patients and that Th17 polarisation could be rescued with the addition of NaCl to culture conditions. **This raises the possibility that better correction of extracellular ionic concentrations *in vivo* in SLT patients may improve IL-17 responses and mitigate infection risk and warrants further study. Moreover, our data provide the basis for investigating whether manipulation of the extracellular ionic environment may be used as a therapeutic strategy in IL-17 mediated inflammatory diseases.”**

REVIEWERS' COMMENTS:

Reviewer #2 (Remarks to the Author):

The authors have addressed this reviewer's concerns.

Reviewer #3 (Remarks to the Author):

The authors have satisfactorily addressed all questions and comments I had.